# Pass@K Policy Optimization:
# Solving Harder Reinforcement Learning Problems

**Christian Walder & Deep Karkhanis**
Google DeepMind
cwalder/dkarkhanis@google.com

## Abstract

Reinforcement Learning algorithms commonly sample multiple ($n > 1$) solution attempts for each problem and reward them independently. This optimizes for pass@1 performance and prioritizes individual sample performance over the diversity and collective utility of a set of samples. Such algorithms under-utilize the sampling capacity, limiting exploration and eventual improvement on harder examples. As a fix, we propose *Pass-at-k Policy Optimization* (PKPO), a multi-variate transformation on batches of rewards which leads to direct optimization of pass@k performance, thus optimizing for sets of samples that feature a large maximum reward when considered jointly. Our primary contribution is to derive novel low variance unbiased estimators for the pass@k and its gradient, in both the binary and continuous reward settings. We show that optimizing with these estimators reduces to reinforcement learning with (batches of) rewards that have been jointly transformed by a function that is stable and efficient to compute.

While previous efforts propose transformations for $k = n$, our transformations are the first to enable robust optimization of the pass@k for any arbitrary $k \leq n$. Rather than simply trading off pass@1 performance for pass@k gains, our method allows annealing $k$ during training, optimizing both metrics and often achieving strong pass@1 performance alongside significant pass@k gains.

We validate our transformations on illustrative toy experiments, which reveal the variance reducing properties of our formulations. We also include real-world examples using the open-source models GEMMA2 and LLAMA3.1. We find that our transformation effectively optimizes for the target $k$. Furthermore, higher $k$ values enable solving more and harder problems, while annealing $k$ boosts both the pass@1 and pass@k. Crucially, for challenging task sets where conventional pass@1 optimization stalls, our pass@k approach unblocks learning, likely by improving exploration through the prioritization of joint utility over the utility of individual samples

## 1  Introduction

Recent years have seen the rapid rise of large language models (LLMs) trained with internet-scale pretraining data [RNS+18] with post training using both supervised fine-tuning [WBZ+21] and reinforcement learning (RL) [AAA+23, Tea23, Ant, GYZ+25]. The seminal paradigm of RL with human feedback [CLB+17] is limited by the human-derived data it is based on and the reward hacking issues that arise from the use of subjective signals more generally [ABC+21]. To enable progress toward superhuman capabilities, current work is focusing on grounded reward signals that are free of fine-grained human input as in code generation [SJTR23, LWG+22, DLJ+24, YTC+23, GZC+24] and mathematics [LCC+22, AT, CTO+25, YSG+23].

39th Conference on Neural Information Processing Systems (NeurIPS 2025).

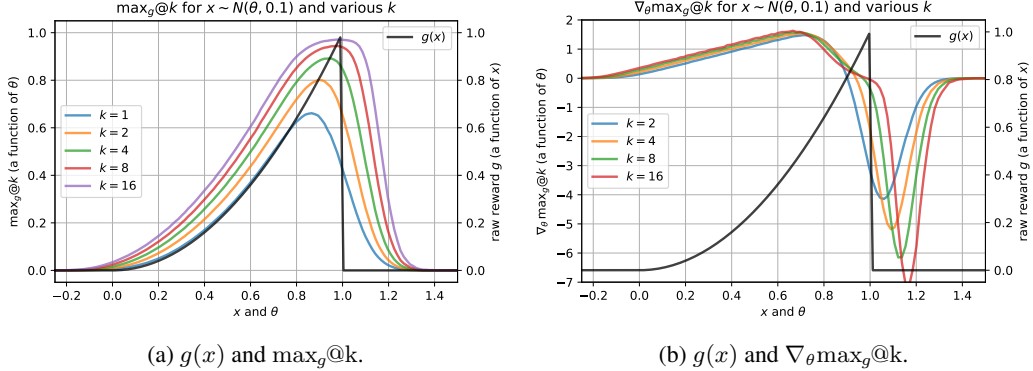

(a) $g(x)$ and $\max_g@k$.        (b) $g(x)$ and $\nabla_\theta \max_g@k$.

Figure 1: The effect of $k$ on the optimal policy for a one-dimensional toy problem. The policy is normal with mean parameter $\theta$ and fixed standard deviation 0.1. For the $\max_g@k$ objective (left, defined in Equation (11)) the optimal $\theta$ corresponds to the horizontal position with maximum $\max_g@k$. For the derivative (right, the estimation of which is the focus of this paper), the optimal $\theta$ corresponds to the location of the zero crossing. For larger $k$ the optimal $\theta$ is more risk tolerant, allowing more samples to exceed one (getting zero reward) in order to increase the chance of obtaining at least one sample close to, but less than one (getting a large reward). See Section 5.1 for more details.

The policy gradients family of RL methods [Wil92] has proven effective in language model training [GYZ+25]. To scale to new capabilities, model training needs to tackle challenging RL task sets with no known solutions but for which correctness may be verified, as in formal mathematics environments [AT]. In such settings, the RL training loop both updates model parameters and searches for solutions to problems at the continuously advancing frontier of model capabilities.

The specific search algorithm introduces a coupling between inference and model updates, which means that naively optimizing the expected single sample reward, or pass@1 may be suboptimal. While various inference-time search methods are possible [HYM+24, KZA+24, LKB+23, WSL+24], simply taking multiple independent samples from the model has proven rather effective [OIW+23]. Our contribution is to couple this simple search method with model parameter updates by enabling robust optimization of the pass@k objective, which is the expected maximum reward over $k$ independent samples.

**Related Literature** The pass@k was championed by [CTJ+21a] who gave a popular unbiased estimator of the metric which we derive from a new perspective (to set up our gradient estimators) as Theorem 1, generalize to continuous rewards in Theorem 3, and provide additional characterisations in Corollaries 2 and 3. Concurrently with our work [TZSM25] offered an elegant variance reduction method for the gradient of the pass@k which corresponds to the special case $n = k$ of our Equation (33). Interpreting pass@k in terms of a partial sort, [CTV19] and [XDC+20] present elegant approximations that are rather general but less efficient in our setting. Others have provided variational approximations for handling the closely related *Best-of-N* [CTG+24, AVAC24] and other more general [BSB+24] inference-time algorithms. The contrasting idea of training a model to approximate the *Best-of-N* prediction with a single sample was addressed by [SDH+24]. Our contribution can be interpreted as a generalization and variance reduction of [TZSM25]. For a general discussion of gradient estimation, variance reduction, and Monte Carlo, we recommend [MRFM20, Owe13].

**Overview and Contributions** Our theme is constructing robust estimators of the pass@k and its gradient given $n \geq k$ samples by averaging (over all $\binom{n}{k}$ subsets of size $k$) simple estimators that are functions of $k$ samples. This is straightforward for binary rewards (Section 2), using the counting proof of Theorem 4. We generalize to continuous rewards using the key trick of assuming without loss of generality that the rewards are sorted, as in Section 3. Finally, we give baselining methods that require more involved derivations due to averaging over all subsets that do not include a given element (to retain unbiasedness) but which boil down to the same easy-to-apply results in Section 4, yielding our *Pass-at-k Policy Optimization* (PKPO). We present toy experiments in Section 5.1 which demonstrate the variance reduction afforded by our estimators. Finally, Section 5.2 demonstrates

that using our reward transformation solves more tasks and selectively optimizes pass@k through RL experiments on GEMMA2 [TRP+24] and LLAMA3.1 [GDJ+24], showcasing real-world impact.

**How to Apply this Method** It is easy to adapt any policy gradient algorithm to use our results. Assume a vector $(g(x_1), g(x_2), \ldots, g(x_n))^\top$ of per-sample rewards for a given task. For example, the $x_i$ could be model samples of source code addressing a specific task (which should be the same for all $n$ samples), and $g$ could provide a numeric score that measures how many tests the code passes, or an overall binary pass indicator, or some combination with additional stylistic or brevity terms, *etc.* Then in order to optimize the pass@k of Equation (1) (or the continuous analog $\max_g$@k of Equation (11)) we simply transform the vector of rewards using either the `sloo` or the `sloo_minus_one` function of Listing 1, which map $\mathbb{R}^n \mapsto \mathbb{R}^n$.[1]

## 2 Binary Rewards

Given a binary reward function $f : \mathcal{X} \to \{0, 1\}$ on the action space $\mathcal{X}$, the pass@k for the model $p(x|\theta)$ is the probability that at least one of $k$ samples drawn i.i.d. is correct:

$$\text{pass@k} = \mathbb{P}\left[\bigvee_{i=1}^{k} [f(x_i) = 1]\right] \tag{1}$$

$$= \mathbb{E}\left[1 - \prod_{i=1}^{k}(1 - f(x_i))\right], \tag{2}$$

where the expectation is over i.i.d. $x_1, x_2, \cdots, x_k \sim p(x|\theta)$.

### 2.1 An Unbiased pass@k Estimator

An estimator for the pass@k was given in [CTJ+21a]: given $n \geq k$ i.i.d. samples of which $c$ are correct, the estimator is

$$\rho(n, c, k) \equiv 1 - \frac{\binom{n-c}{k}}{\binom{n}{k}}. \tag{3}$$

The following was proven in [CTJ+21a]; we give a different proof that sets up our gradient estimator.

**Theorem 1.** $\rho(n, c, k)$ *is an unbiased estimator of the* pass@k.

*Proof.* Let $x_1, x_2, \cdots, x_n \sim p(x|\theta)$, $f_i = f(x_i)$, and $\mathcal{I}$ be a set of $k$ elements sampled uniformly without replacement from $\{1, 2, \ldots, n\}$. Then

$$\text{pass@k} = \mathbb{E}_{x_1, x_2, \ldots, x_n} \mathbb{E}_{\mathcal{I}}\left[1 - \prod_{i \in \mathcal{I}}(1 - f_i)\right]. \tag{4}$$

Averaging over all subsets of size $k$ recovers $\rho$:

$$\frac{1}{\binom{n}{k}} \sum_{\substack{|\mathcal{I}|=k \\ \mathcal{I} \subseteq \{1,2,\ldots,n\}}} \left(1 - \prod_{i \in \mathcal{I}}(1 - f_i)\right) = 1 - \frac{1}{\binom{n}{k}} \sum_{\substack{|\mathcal{I}|=k \\ \mathcal{I} \subseteq \{1,2,\ldots,n\}}} \prod_{i \in \mathcal{I}}(1 - f_i) \tag{5}$$

$$= 1 - \frac{\binom{n-c}{k}}{\binom{n}{k}} \tag{6}$$

$$\equiv \rho(n, c, k), \tag{7}$$

where (6) holds because the sum on the r.h.s. of (5) is the number of subsets of size $k$ of the $(n - c)$ incorrect elements. Since averaging in this way retains unbiasedness, this completes the proof. $\square$

We show in Corollary 2 that no such unbiased estimator exists for $n < k$, and in Corollary 3 that the asymptotic variance of this estimator decreases at a rate of $1/n$.

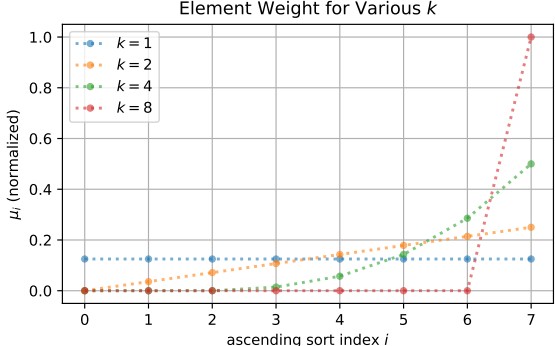

Figure 2: The effect $k$ has on the effective weight $\mu_i / \binom{n}{k}$ of (12) for a mini-batch of size $n = 8$. This is the weight of the contribution of each sample assuming that the samples have been sorted in ascending order from left to right. The horizontal axis is the sort index. For $k = n = 8$ only the largest sample is included; for $k = 1$ all samples are weighted equally. Intermediate values interpolate these extremes in a precise manner that gives rise to unbiased gradient estimation.

## 2.2 An Unbiased pass@k Gradient Estimator

Given a mini-batch of $n$ i.i.d. samples $x_1, x_2, \ldots, x_n$ from $p(x|\theta)$ with corresponding correctness labels $f_i \in \{0, 1\}$, we want to optimize the pass@k w.r.t. the model parameters $\theta$. Letting $c = \sum_{i=1}^n f_i$ be the number of correct samples, we will demonstrate unbiasedness of the estimator

$$\widehat{\nabla} = \sum_{i=1}^n r_i \nabla_\theta \log p(x_i|\theta), \quad \text{where} \quad r_i = \begin{cases} \frac{k}{n} & \text{if } f_i = 1 \\ \frac{k}{n} \cdot \rho(n-1, c, k-1) & \text{if } f_i = 0, \end{cases} \quad (8)$$

that assigns more weight to correct samples, while also assigning some reward to incorrect samples to encourage exploration. The following well-known results will be used to show that (8) is unbiased.

**Lemma 1** (Policy Gradients). *For any absolutely continuous distribution $p(x|\theta)$*

$$\mathbb{E}_{x \sim p(x|\theta)} \left[ r(x) \nabla_\theta \log p(x|\theta) \right] = \nabla_\theta \mathbb{E}_{x \sim p(x|\theta)} \left[ r(x) \right]. \quad (9)$$

**Corollary 1.** *If $c$ is constant w.r.t. both $\theta$ and $x$ then $\mathbb{E}_{p(x|\theta)} \left[ c \nabla_\theta \log p(x|\theta) \right] = 0$.*

*Proof.* By Lemma 1, $\mathbb{E}_{p(x|\theta)} \left[ c \nabla_\theta \log p(x|\theta) \right] = \nabla_\theta \mathbb{E} \left[ c \right] = \nabla_\theta c = 0$. $\qquad\square$

We can now give our first main result:

**Theorem 2.** *$\widehat{\nabla}$ is an unbiased estimator of the gradient of the pass@k:*

$$\mathbb{E}_{x_1, x_2, \ldots, x_n \sim p(x|\theta)} \left[ \widehat{\nabla} \right] = \nabla_\theta \text{pass@k}. \quad (10)$$

See Appendix A.3 for a proof.

## 3 Continuous Rewards

We generalize the pass@k to non-binary rewards $g : \mathcal{X} \to \mathbb{R}$ as

$$\max_g@k \equiv \mathbb{E} \left[ \max \left( \{ g(x_i) \}_{i=1}^k \right) \right]. \quad (11)$$

---

[1]While our experiments focus on `sloo_minus_one`, we recommend experimenting with both estimators.

### 3.1 An Unbiased $\max_g$@k Estimator

The following estimator for the $\max_g$@k is a direct analog of $\rho$: given $n \geq k$ i.i.d. samples, assuming w.l.o.g. that the rewards $g_i = g(x_i)$ are sorted, so that $g_1 \leq g_2 \leq \cdots \leq g_n$ the estimator is

$$\rho^{(g)}(n, c, k) \equiv \frac{1}{\binom{n}{k}} \sum_{i=k}^{n} \mu_i g_i, \tag{12}$$

where

$$\mu_i = \binom{i-1}{k-1}. \tag{13}$$

To compute this stably we cancel factors in the binomial coefficients to get[2]

$$\rho^{(g)}(n, c, k) \equiv \frac{k}{n-k+1} \sum_{i=k}^{n} g_i \prod_{j=1}^{k-1} \frac{i-j}{n-j+1}. \tag{14}$$

**Theorem 3.** $\rho^{(g)}(n, c, k)$ is an unbiased estimator of the $\max_g$@k.

*Proof.* The proof is similar to Theorem 1. Here we exploit the assumption that the $g_i$ are sorted, so

$$\frac{1}{\binom{n}{k}} \sum_{\substack{|\mathcal{I}|=k \\ \mathcal{I} \subseteq \{1,2,\ldots,n\}}} \max_{i \in \mathcal{I}} g_i = \frac{1}{\binom{n}{k}} \sum_{\substack{|\mathcal{I}|=k \\ \mathcal{I} \subseteq \{1,2,\ldots,n\}}} g_{\max_{i \in \mathcal{I}}} \tag{15}$$

$$= \frac{1}{\binom{n}{k}} \sum_{i=k}^{n} \mu_i g_i \tag{16}$$

$$\equiv \rho^{(g)}(n, c, k), \tag{17}$$

since $\mu_i$ is the number of subsets of $1, 2, \ldots, i-1$ of size $k-1$, which equals $\binom{i-1}{k-1}$. The sum starts at $k$ because all subsets of size $k$ include elements that are greater than or equal to $g_k$. $\square$

See Line 17 of Listing 1 for an implementation of $\rho^{(g)}$.

### 3.2 An Unbiased $\max_g$@k Gradient Estimator

We propose the gradient estimator

$$\widehat{\nabla}^{(g)} = \sum_{i=1}^{n} s_i \nabla_\theta \log p(x_i|\theta), \tag{18}$$

where if we assume w.l.o.g. that the $g_i$ are sorted, the $s_i$ are a weighted combination of them,

$$s_i = \frac{1}{\binom{n}{k}} \sum_{j=i}^{n} m_{ij} g_j, \tag{19}$$

where the diagonals are

$$m_{ii} = \begin{cases} \binom{i-1}{k-1} & \text{if } i \geq k-1 \\ 0 & \text{otherwise,} \end{cases} \tag{20}$$

and the off-diagonals are

$$m_{ij} = \begin{cases} \binom{j-2}{k-2} & \text{if } (j > i) \wedge (j \geq k) \wedge (k \geq 2) \\ 0 & \text{otherwise.} \end{cases} \tag{21}$$

**Theorem 4.** $\widehat{\nabla}^{(g)}$ is an unbiased estimator of the gradient of the $\max_g$@k:

$$\mathbb{E}_{x_1, x_2, \ldots, x_n \sim p(x|\theta)} \left[ \widehat{\nabla}^{(g)} \right] = \nabla_\theta \max_g\text{@k.} \tag{22}$$

---

[2]We thank to Ruixu Zhou of Tsinghua University for correcting errors in equations 14, 31 and 32.

*Proof.* The proof is analogous to that of Theorem 2. Here we have

$$\widehat{\nabla} \equiv \rho^{(g)}(n,c,k)\nabla_\theta \sum_{i=1}^n \log p(x_i|\theta) \tag{23}$$

$$= \frac{1}{\binom{n}{k}} \sum_{\substack{|\mathcal{I}|=k \\ \mathcal{I} \subseteq \{1,2,\dots,n\}}} \max_{j \in \mathcal{I}} g_j \sum_{i=1}^n \nabla_\theta \log p(x_i|\theta) \tag{24}$$

$$\overset{\mathbb{E}}{\equiv} \frac{1}{\binom{n}{k}} \sum_{i=1}^n \nabla_\theta \log p(x_i|\theta) \sum_{j=1}^n m_{ij} g_j, \tag{25}$$

By assumption the $g_i$ are sorted, so $\max_{j \in \mathcal{I}} g_j = g_{\max_{j \in \mathcal{I}}}$. Therefore $m_{ij}$ is the number of subsets $\mathcal{I}$ of $\{1, 2, \dots, n\}$ that

1. are of size $k$,

2. have $j \geq i$ as the largest element (so that we can factor out $g_j$),

3. include $i$ (so that (25) holds in expectation by Corollary 1).

Due to the second condition, the form of $m_{ij}$ depends on whether $i = j$.

The diagonals $m_{ii}$ are zero if $i < k$ since the largest element of any subset of size $k$ is at least $k$. If $i \geq k$ then we fix $i$ and are left with $i - 1$ elements from which to choose $k - 1$ which we can do $\binom{i-1}{k-1}$ ways in line with (20).

The $m_{ij}$ for $i \neq j$ are obtained by fixing $i$ and $j$ leaving $j - 2$ elements $1, 2, \dots, i - 1, \dots, i + 1, \dots, j - 1$ from which to choose $k - 2$ which we can do $\binom{j-2}{k-2}$ ways in line with (21). $\qquad\square$

**Theorem 5.** $s_1, s_2, \dots, s_n$ *can be computed in total time* $\mathcal{O}(k + n \log n)$.

See Appendix A.4 for the proof and Line 36 of Listing 1 for an implementation based on it.

## 4  Variance Reduction

### 4.1  Leave-One-Out Baseline for the Simple Case

A popular variance reduction method [MRFM20, Owe13, GYZ⁺25] for point-wise rewards $g(x)$ subtracts the mean of the leave one out (LOO) rewards within each mini-batch $x_1, x_2, \dots, x_n$:

$$g^{(\mathrm{loo})}(x_i) = g(x_i) - \frac{1}{n-1} \sum_{\substack{j=1 \\ j \neq i}}^n g(x_j). \tag{26}$$

Since the subtracted part does not depend on $x_i$, by Corollary 1 this retains unbiasedness.

### 4.2  Leave-One-Out Baseline for $\max_g$@k

Baselining the $s_i$ of (19) in this way introduces bias, however, as each $s_i$ depends on all $x_1, \dots, x_n$. We instead apply LOO to the following form of $s_i$ that follows from Theorem 4 and the proof thereof:

$$s_i = \frac{1}{\binom{n}{k}} \sum_{\substack{|\mathcal{I}|=k \\ i \in \mathcal{I} \\ \mathcal{I} \subseteq \{1,2,\dots,n\}}} \max_{j \in \mathcal{I}} g_j \tag{27}$$

$$\equiv S(i, k, \{1, 2, \dots, n\}), \tag{28}$$

We can then retain unbiasedness by excluding $i$ from the baseline, by defining

$$s_i^{(\mathrm{loo})} \equiv S(i, k, \{1, 2, \dots, n\}) - \frac{1}{n-1} \sum_{\substack{j=1 \\ j \neq i}}^n S(j, k, \{1, 2, \dots, n\} \setminus i). \tag{29}$$

**Theorem 6.** $s_1^{(loo)}, s_2^{(loo)}, \ldots, s_n^{(loo)}$ *can be computed in total time* $\mathcal{O}(k + n \log n)$.

*Proof.* Given (5) it is sufficient to consider computing, for $i = 1, 2, \ldots, n$,

$$b_i^{(k)} \equiv \sum_{\substack{j=1 \\ j \neq i}}^{n} S(j, k, \{1, 2, \ldots, n\} \setminus i). \tag{30}$$

By assuming w.l.o.g. an ascending ordering of $g(x_i)$, excluding the first index does not change the ordering of the remaining indices. The first term is therefore[2]

$$b_1^{(k)} = \sum_{i=2}^{N} s_i = \frac{1}{\binom{n-1}{k}} \sum_{i=2}^{N} \left( m_{ii} + m_{i-1,i}(i-2) \right) g(x_i), \tag{31}$$

where (31) follows from (49). From (49) we obtain for $1 \leq i < n$ the left to right recursion

$$b_{i+1}^{(k)} = b_i^{(k)} + \frac{1}{\binom{n-1}{k}} \left( g(x_i) - g(x_{i+1}) \right) \left( m_{ii} + m_{i-1,i}(i-2) \right). \tag{32}$$

Similar arguments to the proof of Theorem 5 therefore imply the same time complexity. $\square$

Line 52 of Listing 1 implements $s_i^{(loo)}$ using the recursion in the above proof.

### 4.3 $\max_g @(\mathrm{k} - 1)$ **Leave-One-Out Baseline for** $\max_g @\mathrm{k}$

The baseline $b_i^{(k)}$ is an average of the $\max_g @\mathrm{k}$ estimates over sets of size $k$. For a number of samples $n$ equal to $k$, there are no such subsets to construct the baseline. [TZSM25] recently overcame this issue for the specific case $n = k$ by using $\max_g @(\mathrm{k} - 1)$ as the baseline statistic. We generalize their approach to $k < n$ and to averaging over all subsets by defining similarly to Equation (27)

$$s_i^{(loo-1)} = \frac{1}{\binom{n}{k}} \sum_{\substack{|\mathcal{I}|=k \\ i \in \mathcal{I} \\ \mathcal{I} \subseteq \{1,2,\ldots,n\}}} \left( \max_{j \in \mathcal{I}} g_j - \max_{b \in \mathcal{I} \setminus i} g_b \right). \tag{33}$$

Averaging smaller but more numerous subsets in the baseline reduces variance but introduces bias (in the baseline, not $s_i^{(loo-1)}$). Given our previous results it is straightforward to show

**Theorem 7.** $s_1^{(loo-1)}, s_2^{(loo-1)}, \ldots, s_n^{(loo-1)}$ *can be computed in total time* $\mathcal{O}(k + n \log n)$.

*Proof.* By the linearity of the expectation we can split the two terms in the parentheses of Equation (33) into two separate sums. The first summation is by definition simply $s_i$ of Equation (19). The (negation of the) second summation can be computed efficiently using

$$\frac{1}{\binom{n}{k}} \sum_{\substack{|\mathcal{J}|=k \\ \mathcal{J} \subseteq \{1,2,\ldots,n\}}} \max_{b \in \mathcal{J} \setminus i} g_b = \frac{1}{\binom{n}{k}} \sum_{\substack{|\mathcal{B}|=k-1 \\ \mathcal{B} \subseteq \{1,2,\ldots,n\} \setminus i}} \max_{b \in \mathcal{B}} g_b = \frac{k}{n(k-1)} b_i^{(k-1)}, \tag{34}$$

where the final equality follows with a little algebra from Equation (28) and Equation (30). $\square$

Listing 1 implements $s_i^{(loo-1)}$ using (34); Figure 5 compares $s_i$, $s_i^{(loo)}$, and $s_i^{(loo-1)}$.

## 5 Experiments

### 5.1 One-Dimensional Toy Example

We start with a policy that is Gaussian with a fixed standard deviation and mean parameter $\theta$ we wish to learn, so that $x \sim \mathcal{N}(\theta, 0.1)$. We set the raw reward to be

$$g(x) = \begin{cases} x^2 & 0 \leq x \leq 1 \\ 0 & \text{otherwise.} \end{cases} \tag{35}$$

The optimal policy under the $\max_g @\mathrm{k}$ reward varies with $k$ (see Figure 1). The variance of our estimators is compared in Figure 4 where $s^{(loo-1)}$ is the strongest.

Table 1: Results for GEMMA2-9B on the MATH benchmark.

| GEMMA2-9B | k_eval=1 | k_eval=2 | k_eval=4 | k_eval=8 | k_eval=16 |
|---|---|---|---|---|---|
| k_opt=1 | **22.24** $\pm$ 0.50 | 25.35 $\pm$ 0.55 | 30.73 $\pm$ 0.59 | 37.08 $\pm$ 0.64 | 42.59 $\pm$ 0.68 |
| k_opt=2 | 21.46 $\pm$ 0.51 | **28.61** $\pm$ 0.56 | 32.92 $\pm$ 0.61 | 39.59 $\pm$ 0.66 | 45.34 $\pm$ 0.70 |
| k_opt=4 | 21.25 $\pm$ 0.53 | 27.15 $\pm$ 0.58 | **34.93** $\pm$ 0.63 | 41.71 $\pm$ 0.69 | 47.05 $\pm$ 0.74 |
| k_opt=8 | 20.69 $\pm$ 0.56 | 26.78 $\pm$ 0.60 | 33.68 $\pm$ 0.66 | **42.62** $\pm$ 0.72 | **48.37** $\pm$ 0.77 |
| [TZSM25] | 19.48 $\pm$ 0.61 | 25.41 $\pm$ 0.67 | 31.17 $\pm$ 0.73 | 39.34 $\pm$ 0.79 | 44.82 $\pm$ 0.83 |
| EntropyReg | 20.85 $\pm$ 0.58 | 26.05 $\pm$ 0.64 | 32.48 $\pm$ 0.70 | 38.21 $\pm$ 0.76 | 43.95 $\pm$ 0.81 |

Table 2: Results for LLAMA3.1-8B on the MATH benchmark.

| LLAMA3.1-8B | k_eval=1 | k_eval=2 | k_eval=4 | k_eval=8 | k_eval=16 |
|---|---|---|---|---|---|
| k_opt=1 | **51.15** $\pm$ 0.61 | 51.82 $\pm$ 0.64 | 53.69 $\pm$ 0.68 | 55.41 $\pm$ 0.72 | 56.83 $\pm$ 0.76 |
| k_opt=2 | 49.72 $\pm$ 0.62 | **53.51** $\pm$ 0.66 | 55.45 $\pm$ 0.70 | 57.23 $\pm$ 0.74 | 58.71 $\pm$ 0.78 |
| k_opt=4 | 49.18 $\pm$ 0.64 | 52.20 $\pm$ 0.68 | **57.83** $\pm$ 0.72 | 58.47 $\pm$ 0.77 | 59.28 $\pm$ 0.81 |
| k_opt=8 | 48.63 $\pm$ 0.67 | 52.14 $\pm$ 0.71 | 56.28 $\pm$ 0.75 | **59.04** $\pm$ 0.80 | **61.88** $\pm$ 0.84 |
| [TZSM25] | 48.21 $\pm$ 0.70 | 50.93 $\pm$ 0.75 | 54.38 $\pm$ 0.80 | 57.11 $\pm$ 0.85 | 58.55 $\pm$ 0.90 |
| EntropyReg | 48.51 $\pm$ 0.68 | 51.95 $\pm$ 0.73 | 55.33 $\pm$ 0.78 | 56.95 $\pm$ 0.83 | 58.18 $\pm$ 0.88 |

Table 3: Results for GEMMA2-9B on the Coding benchmark.

| GEMMA2-9B | k_eval=1 | k_eval=2 | k_eval=4 | k_eval=8 | k_eval=16 |
|---|---|---|---|---|---|
| k_opt=1 | **37.71** $\pm$ 0.60 | 42.03 $\pm$ 0.65 | 48.19 $\pm$ 0.69 | 55.07 $\pm$ 0.75 | 60.98 $\pm$ 0.79 |
| k_opt=2 | 36.84 $\pm$ 0.61 | **46.56** $\pm$ 0.67 | 52.68 $\pm$ 0.72 | 59.73 $\pm$ 0.78 | 65.86 $\pm$ 0.84 |
| k_opt=4 | 36.49 $\pm$ 0.63 | 44.95 $\pm$ 0.69 | **57.09** $\pm$ 0.76 | 63.64 $\pm$ 0.83 | 69.51 $\pm$ 0.88 |
| k_opt=8 | 35.75 $\pm$ 0.67 | 44.41 $\pm$ 0.73 | 55.08 $\pm$ 0.80 | **65.56** $\pm$ 0.88 | **71.91** $\pm$ 0.94 |
| [TZSM25] | 34.36 $\pm$ 0.72 | 42.81 $\pm$ 0.78 | 52.36 $\pm$ 0.86 | 61.07 $\pm$ 0.95 | 66.41 $\pm$ 1.01 |
| EntropyReg | 35.91 $\pm$ 0.70 | 43.75 $\pm$ 0.76 | 53.28 $\pm$ 0.84 | 60.13 $\pm$ 0.93 | 65.29 $\pm$ 0.99 |

Table 4: Results for LLAMA3.1-8B on the Coding benchmark.

| LLAMA3.1-8B | k_eval=1 | k_eval=2 | k_eval=4 | k_eval=8 | k_eval=16 |
|---|---|---|---|---|---|
| k_opt=1 | **67.38** $\pm$ 0.72 | 67.45 $\pm$ 0.76 | 69.22 $\pm$ 0.80 | 71.11 $\pm$ 0.84 | 72.84 $\pm$ 0.88 |
| k_opt=2 | 64.91 $\pm$ 0.73 | **69.73** $\pm$ 0.78 | 72.03 $\pm$ 0.82 | 74.08 $\pm$ 0.87 | 75.89 $\pm$ 0.91 |
| k_opt=4 | 64.25 $\pm$ 0.75 | 68.47 $\pm$ 0.80 | **74.67** $\pm$ 0.85 | 75.01 $\pm$ 0.90 | 77.75 $\pm$ 0.95 |
| k_opt=8 | 63.57 $\pm$ 0.78 | 68.39 $\pm$ 0.83 | 72.84 $\pm$ 0.88 | **76.82** $\pm$ 0.94 | **79.33** $\pm$ 0.99 |
| [TZSM25] | 62.77 $\pm$ 0.82 | 66.86 $\pm$ 0.88 | 70.83 $\pm$ 0.94 | 73.95 $\pm$ 1.01 | 75.47 $\pm$ 1.06 |
| EntropyReg | 63.91 $\pm$ 0.80 | 67.85 $\pm$ 0.86 | 71.78 $\pm$ 0.92 | 72.99 $\pm$ 0.98 | 74.31 $\pm$ 1.04 |

## 5.2 RL on Open Source LLMs

We demonstrate promising RL results with the 2B and 9B parameter variants of GEMMA2 [TRP+24] and the 8B parameter variant of LLAMA3.1 on real-world problems in MATH [HBK+21], code generation [AON+21] [CTJ+21b], and the easy public subset of ARC-AGI-1 [CKKL25]. The latter is a challenging reasoning task-set even for state-of-the-art models much larger than GEMMA2.

For GEMMA2-2B we use a v5litepod-128 [Goo] which needs around 4 hours per 1000 training steps. Each RL training run [SWD+17] involves sampling a fixed $n$ number of completions $\{x_i\}_{i=1}^n$ for a given prompt at a given training step. For our experiments, we set $n = 16$. The rewards are computed for every completion using a reward function $g(\cdot)$. We transform these rewards $\{g(x_i)\}_{i=1}^n$ using our unbiased estimator $s^{(\text{loo}-1)}$ of (33), which we favour due to Figure 4, and which we refer to as PKPO. We repeat the training for a selection of $k^{\text{opt}}$, thus optimizing a different pass@$k^{\text{opt}}$ each time. Since $k^{\text{opt}} = 1$ leads to no reward transformation, this is our baseline (al-

Table 5: Results for GEMMA2-9B on the `ARC-AGI-1` benchmark.

| GEMMA2-9B | Cumulative Solve Rate | pass@1 | pass@16 |
|---|---|---|---|
| `k_opt=1` | $12.00 \pm 04.33$ | $02.00 \pm 01.69$ | $08.18 \pm 04.00$ |
| `k_opt=4` | $\mathbf{82.33} \pm 04.14$ | $\mathbf{22.00} \pm 02.00$ | $\mathbf{38.18} \pm 04.67$ |
| `k_opt=8` | $\mathbf{84.14} \pm 04.67$ | $\mathbf{26.67} \pm 02.50$ | $\mathbf{44.50} \pm 04.33$ |
| [TZSM25] | $22.00 \pm 04.44$ | $06.00 \pm 02.67$ | $10.16 \pm 04.57$ |
| `EntropyReg` | $24.67 \pm 04.50$ | $04.00 \pm 02.33$ | $08.89 \pm 04.89$ |

Table 6: Results for LLAMA3.1-8B on the `ARC-AGI-1` benchmark.

| LLAMA3.1-8B | Cumulative Solve Rate | pass@1 | pass@16 |
|---|---|---|---|
| `k_opt=1` | $22.00 \pm 04.18$ | $03.33 \pm 02.00$ | $08.00 \pm 04.50$ |
| `k_opt=4` | $\mathbf{87.17} \pm 04.14$ | $\mathbf{24.33} \pm 02.33$ | $\mathbf{42.00} \pm 04.16$ |
| `k_opt=8` | $\mathbf{88.89} \pm 04.33$ | $\mathbf{29.67} \pm 02.67$ | $\mathbf{43.13} \pm 04.67$ |
| [TZSM25] | $36.00 \pm 02.50$ | $08.00 \pm 04.00$ | $18.00 \pm 04.89$ |
| `EntropyReg` | $28.00 \pm 04.44$ | $08.00 \pm 02.50$ | $14.67 \pm 04.44$ |

though we use basic LOO mean centering of Equation (26), without which the training diverges). For each run, we measure pass@k$^{\text{eval}}$ for every $k^{\text{eval}} \in \{1, 2, 4, 8, 12, 16\}$ at each step. Additionally, we also track model entropy and cumulative solve rate during training. The latter is defined as the fraction of tasks from the task-set for which the model has sampled a correct solution at least once; this is a critical metric that reflects the success of the model's exploration and measures its ability to find novel solutions.

**Entropy regularization baseline** In addition to our `PKPO` and the special case thereof of [TZSM25], we also add the entropy regularization baseline, which is PPO with an additional entropy term in the objective. We give this baseline an arguably unfair advantage by performing a small sweep over the values $0.001, 0.005, 0.01, 0.05, 0.1$ for the `entropy_coefficient` for each (model, benchmark) pair and only report the best result as `EntropyReg`.

### 5.2.1 Choosing $k^{\text{opt}}$ selectively optimizes pass@k$^{\text{eval}}$ and solves more tasks

We use the training split of Hendrycks MATH [HBK+21] which contains 12,000 problems as our task set. Figure 6a shows that a higher $k^{\text{opt}}$ in our transformation leads to a consistently higher cumulative solve rate throughout training, as well as a higher entropy. By optimizing pass@k instead of pass@1, the model appears to better utilize the exploration budget thus finding more solutions.

In Figure 7, we compare pass@k$^{\text{eval}}$ across our runs ($k^{\text{opt}} \in \{1, 4, 8\}$) for various $k^{\text{eval}}$. We find the best pass@k$^{\text{eval}}$ when $k^{\text{opt}} = k^{\text{eval}}$ (or $k^{\text{opt}}$ is closest to $k^{\text{eval}}$ among available $k^{\text{opt}}$). Non-transformed rewards optimize pass@1, leading to sub-optimal pass@k$^{\text{eval}}$ for $k^{\text{eval}} \neq 1$, and the deficit worsens as $k^{\text{eval}}$ increases. Thus, our experiments also demonstrate that setting $k^{\text{opt}} := k^{\text{eval}}$ in our transformation suffices to optimize pass@k$^{\text{eval}}$ for a $k^{\text{eval}} \leq n$. This generalizes the already powerful result of [TZSM25] by alleviating the coupling that restricts to optimizing either pass@n or pass@1. In other words, since RL training of LLMs typically samples a large batch ($n \gg 1$), failing to use our transformation results in sub-optimal pass@k performance, especially for modest values of $k$.

As $k^{\text{opt}} \longrightarrow n$, the variance of our estimator increases as there are fewer subsets in (33) (see Figure 4). We presume this is why 1) gains of $k^{\text{opt}} = 8$ over $k^{\text{opt}} = 4$ are more prominent when $k^{\text{eval}} \in \{12, 16\}$ than when $k^{\text{eval}} = 8$. That is, when $k^{\text{eval}}$ is further away from $k^{\text{opt}} \in \{4, 8\}$ than when it is closer, and 2) the special case $n = k^{\text{opt}}$ of [TZSM25] struggles to optimize the pass@n.

### 5.2.2 `PKPO` robustly improves pass@k on held out evaluations

Tables 1-4 above (and Tables 7-8 in the appendix) present performance on held-out sets for two tasks. We report the mean and standard error based on three runs with different random seeds. For math, we train on the train split and evaluate on the test split of Hendrycks MATH [HBK+21]. To

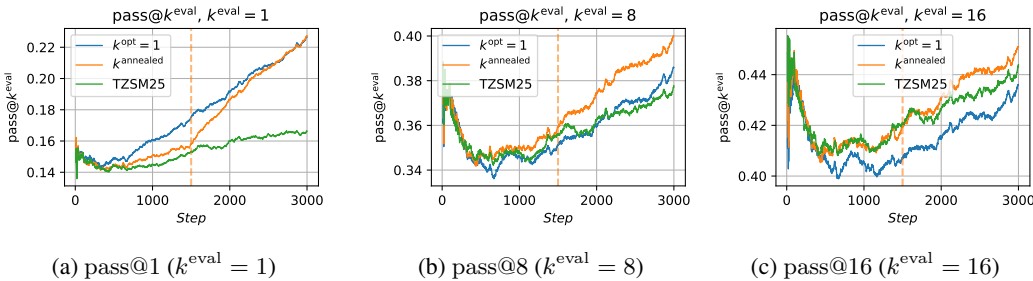

(a) pass@1 ($k^{\mathrm{eval}} = 1$)    (b) pass@8 ($k^{\mathrm{eval}} = 8$)    (c) pass@16 ($k^{\mathrm{eval}} = 16$)

Figure 3: Annealing $k^{\mathrm{opt}}$ during PKPO training improves pass@$k^{\mathrm{eval}}$ without sacrificing pass@1. For $k^{\mathrm{annealed}}$, we train with $k^{\mathrm{opt}} = 8$ up to step 1500 and $k^{\mathrm{opt}} = 1$ thereafter.

evaluate coding, we use MBPP [AON+21] for training and evaluate on HUMANEVAL [CTJ+21b]. MBPP has multiple unit tests per problem and hence we use this not only as a proxy for additional benchmarks but also to showcase our handling of a continuous reward function (% unit tests passed).

### 5.2.3 Improving pass@k without sacrificing pass@1

Figure 3 demonstrates that as PKPO can use any arbitrary $k^{\mathrm{opt}} \leq n$, this allows varying $k^{\mathrm{opt}}$ over the course of training to good effect. We show a simple annealing procedure which starts training with a high $k^{\mathrm{opt}} = 8$ and reduces it to $k^{\mathrm{opt}} = 1$ after 1500 steps. This trains the model to initially prioritize exploration (optimize pass@k) and then consolidate the single-sample policy (optimize pass@1). This switch is apparent in Figure 3a, at step 1500 where the slope of $k^{\mathrm{annealed}}$ changes. While traditional methods like [TZSM25] suffer from a trade-off between pass@k and pass@1, we get a final model which has higher pass@$k^{\mathrm{eval}}$ for all $k^{\mathrm{eval}} > 1$ with no sacrifice in pass@1.

### 5.2.4 PKPO is essential for learning on hard problems

Figure 8 shows the limitation of traditional pass@1 optimization through RL on an especially challenging task-set. We use the easy subset of ARC-AGI-1 [CKKL25]. We observe that conventional pass@1 optimization stalls. However, our pass@k approach unblocks learning, and results in higher pass@$k^{\mathrm{eval}}$ across all $k^{\mathrm{eval}}$ including $k^{\mathrm{eval}} = 1$. Furthermore, we see higher $k^{\mathrm{opt}}$ leads to more effective and faster learning. This is likely because the benefits of prioritizing joint utility over individual sample utility are more prominent on a harder task-set.

Tables 5 and 6 show more extensive experiments on ARC-AGI-1. We make an 80:20 train:test split of the same easy subset as before and report the cumulative solve rate on the train set and pass@k rate on the test set. We train to saturation (no change in cumulative rate for 1k steps), and again use three random restarts to provide standard errors. By encouraging exploration in a direct and stable manner, our method unblocks learning unlike other methods. Entropy Regularization does indeed sacrifice pass@1 and slightly improves pass@k by promoting exploration, but it is hard to tune, and is significantly outperformed by our method. Moreover, it has no explicit way to optimize for a specific k_eval. [TZSM25] targets the same objective as PKPO, but couples the minibatch size to $k$ and thereby incurs higher variance than PKPO with $k < n$.

## 6 Conclusions and Outlook

In RL training with multiple independent samples per task, optimizing the pass@k maximizes the expectation of the *best* reward in the set of samples, rather than the *average* one. This preserves model output diversity, which leads to solving more problems and ultimately yields stronger policies. We provide drop-in replacements for more traditional RL reward transformations that robustly and efficiently optimize the pass@k. This work can be extended in various ways, such as to other inference-time search algorithms, and to more sophisticated baseline techniques.

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

# A   Additional Theoretical Statements and Proofs

## A.1   Statement and proof that $n \geq k$ samples are required to unbiasedly estimate pass@k

This result is a direct consequence of a well-known theorem concerning the unbiased estimability of parametric functions for the Bernoulli distribution.

**Theorem 8** (Kolmogorov [Kol50]). *Let $Y_1, \ldots, Y_n$ be i.i.d. Bernoulli random variables with success probability $p \in [0,1]$. A function $\rho(p)$ is unbiasedly estimable from this sample if and only if it can be expressed as a polynomial in $p$ of degree at most $n$.*

A sketch of a proof of Theorem 8 can be found in Lehmann and Casella [Leh98].

**Corollary 2.** *Given a sequence of $n$ i.i.d. model samples $x_1, x_2, \ldots, x_n$, the pass@k is unbiasedly estimable if and only if $n \geq k$.*

*Proof.* It is sufficient to consider a single a fixed and observed correctness function $f$, so that the independence of the $x_i$ implies the independence of the correctness events $[f(x_i) = 1]$. Let $p = \mathbb{P}\big[[f(x_i) = 1]\big]$ be the probability that any single sample is correct. The pass@k is defined as the complement of the probability that all $k$ samples are incorrect, which for the specific assumptions adopted in this proof is $1 - (1-p)^k$. Because this expression is a polynomial in $p$ of degree $k$, the result follows immediately from Theorem 8. $\square$

## A.2   Characterization of the Variance

Our proof of Theorem 1 identifies the pass@k estimator $\rho(n, c, k)$ as a $U$-statistic. To characterize its variance, we apply Hoeffding's asymptotic theory.

**Theorem 9** (Hoeffding [Hoe48]). *Let $X_1, \ldots, X_n$ be independent and identically distributed random variables with distribution $F$. Let $h(x_1, \ldots, x_k)$ be a symmetric kernel with $\mathbb{E}[h(X_1, \ldots, X_k)^2] < \infty$. Define the parameter $\mu = \mathbb{E}_F[h(X_1, \ldots, X_k)]$ and the $U$-statistic:*

$$U_n = \binom{n}{k}^{-1} \sum_{1 \leq i_1 < \cdots < i_k \leq n} h(X_{i_1}, \ldots, X_{i_k}). \tag{36}$$

*Let $h_1(x) = \mathbb{E}[h(x, X_2, \ldots, X_k)]$ be the projection of the kernel onto a single variable. Hoeffding proved that if $\zeta_1 = Var(h_1(X_1)) > 0$, then as $n \to \infty$:*

$$\sqrt{n}(U_n - \mu) \xrightarrow{d} \mathcal{N}(0, k^2 \zeta_1). \tag{37}$$

In the standard application of pass@k we evaluate the estimator on a specific problem defined by a prompt and a correctness oracle. While the true pass rate $\nu$ is unknown to the observer, it is a fixed property of the model-problem pair. Consequently, the correctness outcomes of the generated samples are i.i.d. conditioned on the problem.

The following lemma derives the variance parameter $\zeta_1$ under this conditioning. We abuse the notation by allowing the $X_i$ to denote correctness.

**Lemma 2** (Conditional Variance of the Projection). *Fix a problem instance such that the correctnesses $X_i$ are i.i.d. Bernoulli($\nu$). For the pass@k kernel $h(x_1, \ldots, x_k) = \max(x_1, \ldots, x_k)$, the variance of the first-order projection is:*

$$\zeta_1(\nu, k) = \nu(1 - \nu)^{2k-1}. \tag{38}$$

*Proof.* The projection $h_1(x)$ is the expected value of the kernel given the first sample is fixed to $x$, while $X_2, \ldots, X_k$ remain random variates drawn from Bernoulli($\nu$).

$$h_1(x) = \mathbb{E}[\max(x, X_2, \ldots, X_k)]. \tag{39}$$

We evaluate this for the two possible realizations of $x$:

1. **Case $x = 1$ (Success):** The maximum is 1 regardless of the remaining samples.
$$h_1(1) = 1.$$

2. **Case $x = 0$ (Failure):** The maximum is 0 if and only if all remaining $k - 1$ samples fail. Since the remaining samples are i.i.d. with failure probability $(1 - \nu)$,
$$h_1(0) = 1 - (1 - \nu)^{k-1}.$$

The projection $h_1(X_1)$ is thus a binary random variable taking value $h_1(1)$ with probability $\nu$ and $h_1(0)$ with probability $1 - \nu$, so that

$$\begin{aligned}
\zeta_1 &= \nu(1 - \nu)\left(h_1(1) - h_1(0)\right)^2 \\
&= \nu(1 - \nu)\left(1 - [1 - (1 - \nu)^{k-1}]\right)^2 \\
&= \nu(1 - \nu)\left((1 - \nu)^{k-1}\right)^2 \\
&= \nu(1 - \nu)^{2k-1}.
\end{aligned}$$

$\square$

We can now substitute this explicit form back into Hoeffding's general result.

**Corollary 3.** *For a fixed problem with pass rate $\nu$, as $n \to \infty$, the asymptotic variance of the estimator $\rho(n, c, k)$ is:*

$$Var(\rho) \approx \frac{1}{n}\left[k^2\nu(1 - \nu)^{2k-1}\right]. \tag{40}$$

### A.3 Proof of Theorem 2

Although Theorem 2 is a special case of Theorem 4, we include both because the following proof uses a different approach from that of the more general statement, and is arguably the easier of the two.

*Proof.* By Lemma 1 the gradient $\nabla_\theta$pass@k has the unbiased estimator

$$\widehat{\nabla} \equiv \rho(n, c, k)\nabla_\theta \sum_{i=1}^{n} \log p(x_i|\theta) \tag{41}$$

$$= \frac{1}{\binom{n}{k}} \sum_{\substack{\mathcal{I} \subseteq \{1,2,\ldots,n\} \\ |\mathcal{I}|=k}} \left(1 - \prod_{i \in \mathcal{I}}(1 - f_i)\right) \sum_{i=1}^{n} \nabla_\theta \log p(x_i|\theta) \tag{42}$$

$$\stackrel{\mathbb{E}}{\equiv} \frac{1}{\binom{n}{k}} \sum_{i=1}^{n} m_i \nabla_\theta \log p(x_i|\theta), \tag{43}$$

where (42) substitutes the l.h.s. of (5). $m_i$ is the number of subsets $\mathcal{I}$ of $\{1, 2, \ldots, n\}$ that

1. are of size $k$,

2. contain at least one correct element, so that $\left(1 - \prod_{i \in \mathcal{I}}(1 - f_i)\right) = 1$,

3. contain $i$, so that (43) holds in expectation by Corollary 1.

Due to the second condition, $m_i$ therefore equals one of two values, which we denote by $m^{(1)}$ and $m^{(0)}$, depending on whether $f_i = 1$ or $f_i = 0$, respectively.

If $f_i = 1$ then all subsets that include $i$ also include at least one correct element ($i$ itself), so that $m^{(1)}$ is just the number of subsets of size $k$ of $\{1, 2, \ldots, n\}$ that include $i$, which equals the number of subsets of size $k - 1$ of $\{1, 2, \ldots, n - 1\}$:

$$m^{(1)} = \binom{n-1}{k-1}. \tag{44}$$

If $f_i = 0$ then we assume w.l.o.g. that $i = n$, so that $m^{(0)}$ is the number of subsets of size $k - 1$ of $\{1, 2, \ldots, n - 1\}$ with at least one correct element,

$$m^{(0)} = \sum_{\substack{\mathcal{J} \subseteq \{1,2,\ldots,n-1\} \\ |\mathcal{J}| = k-1}} \left( 1 - \prod_{j \in \mathcal{J}} (1 - f_j) \right) \equiv \binom{n-1}{k-1} \rho(n-1, c, k-1), \tag{45}$$

where we again used (5), this time to get an expression in terms of $\rho$. Using $m^{(0)}$ and $m^{(1)}$ we can compute $r^{(0)}$ and $r^{(1)}$ using (43) as

$$r^{(1)} = \frac{m^{(1)}}{\binom{n}{k}} = \frac{\binom{n-1}{k-1}}{\binom{n}{k}} = \frac{k}{n}, \tag{46}$$

and

$$r^{(0)} = \frac{m^{(0)}}{\binom{n}{k}} = \frac{\binom{n-1}{k-1} \rho(n-1, c, k-1)}{\binom{n}{k}} = \frac{k}{n} \cdot \rho(n-1, c, k-1), \tag{47}$$

in line with (8). $\qquad \square$

## A.4 Proof of Theorem 5

*Proof.* The vector $\boldsymbol{s} = (s_1, s_2, \ldots, s_n)^\top$ can be written as $\boldsymbol{s} = M\boldsymbol{g}$ where we have introduced $\boldsymbol{g} = (g(x_1), g(x_2), \ldots, g(x_n))^\top$ as well as the matrix $M$ with

1. diagonal elements $m_{ii}$ given by (20),

2. upper diagonals $m_{ij}$ for $i < j$ given by (21) which is independent of $i$,

3. lower diagonals $m_{ij}$ for $i > j$ equal to zero.

Because of the structure of $M$, we have that

$$s_n = \frac{1}{\binom{n}{k}} m_{nn} \, g(x_n), \tag{48}$$

and, for $1 \le i < n$, the right to left recursion

$$s_i = s_{i+1} + \frac{1}{\binom{n}{k}} \Big( g(x_i) m_{ii} + g(x_{i+1}) \big( m_{i,i+1} - m_{i+1,i+1} \big) \Big). \tag{49}$$

The ratios of $m_{ii}, m_{i,i+1}$ and $m_{i+1,i+1}$ divided by $\binom{n}{k}$ can be simplified by cancelling factors in the binomial coefficients and writing the remaining factors as a product of $k$ ratios similarly to (14), for a total cost of $\mathcal{O}(nk)$; this computation can be further simplified by noting that the required ratios can be lazily computed in sequence (for example to obtain $m_{i+1,i+1}$ from $m_{ii}$) at a cost of $\mathcal{O}(1)$ after computing the first at a cost of $O(k)$, giving a total cost of $\mathcal{O}(k+n)$. The additional $\mathcal{O}(n \log n)$ comes from assuming the $i$ are sorted in increasing order of $g(x_i)$. $\qquad \square$

# B Implementation

```python
def _m_normed(N: int, K: int, i: int, j: int) -> float:
  if i == j and i >= K-1:
    return (
        K / (N-K+1) *
        np.prod(np.arange(i-K+2, i+1) / np.arange(N-K+2, N+1))
    )
  elif j > i and j >= K-1 and K >= 2:
    return (
        K / (N-K+1) * (K-1) / N *
        np.prod(np.arange(j-K+2, j) / np.arange(N-K+2, N))
    )
  return 0

def _m_diagonal(N: int, K: int) -> np.ndarray:
  return np.array([_m_normed(N, K, i, i) for i in range(N)])

def rho(g: np.ndarray, K: int) -> float:
  """See Equation (12)."""
  return (np.sort(g) * _m_diagonal(len(g), K)).sum()

def _delta(N: int, K: int, i: int) -> float:
  return _m_normed(N, K, i, i+1) - _m_normed(N, K, i+1, i+1)

def _deltas(N: int, K: int) -> np.ndarray:
  return np.array([_delta(N-1, K, i) for i in range(N-2)])

def _sorted_apply(func: Callable) -> Callable:
  def inner(x: np.ndarray, *args, **kwargs) -> np.ndarray:
    i_sort = np.argsort(x)
    func_x = np.zeros_like(x)
    func_x[i_sort] = func(x[i_sort], *args, **kwargs)
    return func_x
  return inner

@_sorted_apply
def s(g: np.ndarray, K: int):
  """See Equation (19)."""
  N = len(g)
  c = g * _m_diagonal(N, K)
  c[:(N-1)] += g[1:] * _deltas(N+1, K)
  return np.cumsum(c[::-1])[::-1]

@_sorted_apply
def _b(g: np.ndarray, K: int) -> np.ndarray:
  N = len(g)
  w = (_m_diagonal(N-1, K) * np.arange(1, N)).astype(float)
  w[1:] += _deltas(N, K) * np.arange(1, N-1)
  c1 = np.array([(w * g[1:]).sum()])
  c2 = (g[:-1] - g[1:]) * w
  return np.cumsum(np.concatenate((c1, c2)))

def sloo(g: np.ndarray, K: int) -> np.ndarray:
  """See Equation (29)."""
  return s(g, K) - _b(g, K) / (len(g) - 1)

def sloo_minus_one(g: np.ndarray, K: int) -> np.ndarray:
  """See Equation (33)."""
  return s(g, K) - _b(g, K-1) * K / (K-1) / len(g)
```

Listing 1: Python reward batch transformations. Functions with names that begin with an underscore are helpers, while the remaining four functions rho, s, sloo and sloo_minus_one implement $\rho^{(g)}$, $s_i$, $s_i^{(\text{loo})}$ and $s_i^{(\text{loo}-1)}$, respectively. For simplicity this implementation costs $\mathcal{O}(nk + n \log n)$ — reducing this to $\mathcal{O}(k + n \log n)$ would require optimizing _deltas and _m_diagonal.

# C   Additional Figures

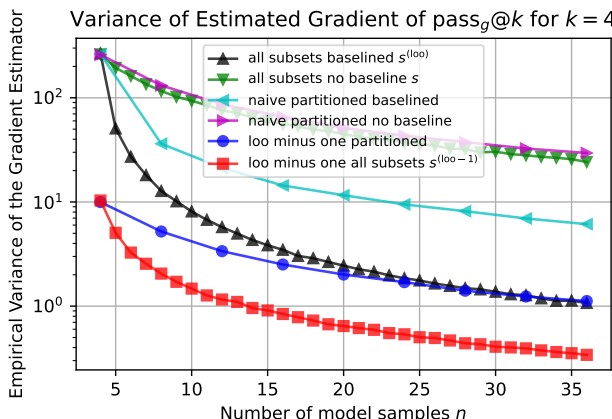

Figure 4: The variance of different estimators of the gradient of $\max_g@\mathrm{k}$ with $k = 4$ for the one-dimensional problem depicted in Figure 1 at the location $x = 1$. Each data-point is the sample variance of 10,000 independent unbiased gradient estimates (lower is better). The horizontal axis denotes the number of samples $n$ used to construct each of the 10,000 estimates. We compare the following methods:

**all subsets baselined:** $s^{(\mathrm{loo})}$ — our novel estimator of Equation (29) that analytically sums over all subsets of size $k$ of the $n$ samples with our unbiased baseline method that subtracts for each element $i$ the mean of the estimator over all subsets of size $k$ that do not include $i$.

**all subsets no baseline:** $s$ — our novel estimator of Equation (19) that analytically sums over all subsets of size $k$ of the $n$ samples but that does not include a variance-reducing baseline.

**naive partitioned baselined** — a naive transformation that sets all $k$ transformed rewards in a subset of $k$ samples equal to the largest raw reward in that subset. To extend this method to $n > k$ we partition the $n$ samples (for integer multiples of $k$) into disjoint subsets of size $k$ and average the estimated gradient obtained from each. Furthermore, as a simple variance reduction method, for each such set of $k$ samples we subtract the mean of the transformed rewards from the other sets of $k$ samples (thereby averaging over $(n - k)$ samples and subtracting the result from the $k$ samples and repeating $n/k$ times in a leave-one-out fashion over the subsets of size $k$). If we were to randomly sample an increasing number of partitions of the samples and average over all of them, then intuitively the resulting estimator would approach the variance of $s^{(\mathrm{loo})}$, but this would be expensive and indeed the limiting case of considering all partitions is intractable for general $n$ and $k$. Our estimators have the key property of summing over all such partitions while nonetheless being efficient to compute.

**naive partitioned no baseline** — a similar method to the previous one, but without the naive mean subtraction based variance reduction step.

**loo minus one partitioned** — a method that uses the same partitioning approach as the previous two, but instead of using the naive estimate (which sets every transformed reward to simple max of the raw reward in a given set of $k$ samples) it uses the $s^{(\mathrm{loo}-1)}$ method applied separately to each disjoint set of $k$ samples, and averages that over all such subsets. In this way, this is a trivial generalization of [TZSM25] which extends to $n > k$ by applying the basic method to disjoint subsets and averaging the results. We do not subtract a baseline across sets as this did not improve the variance, possibly because the method within each $k$ already includes a variance reduction baseline.

**loo minus one all subsets:** $s^{(\mathrm{loo}-1)}$ — our novel estimator of Equation (33) that analytically sums over all subsets of size $k$ of the $n$ samples and uses all appropriate subsets of size $k - 1$ to form the variance-reducing baseline that retains unbiasedness, thereby non-trivially generalizing [TZSM25] to all $n > k$ with strong variance reduction.

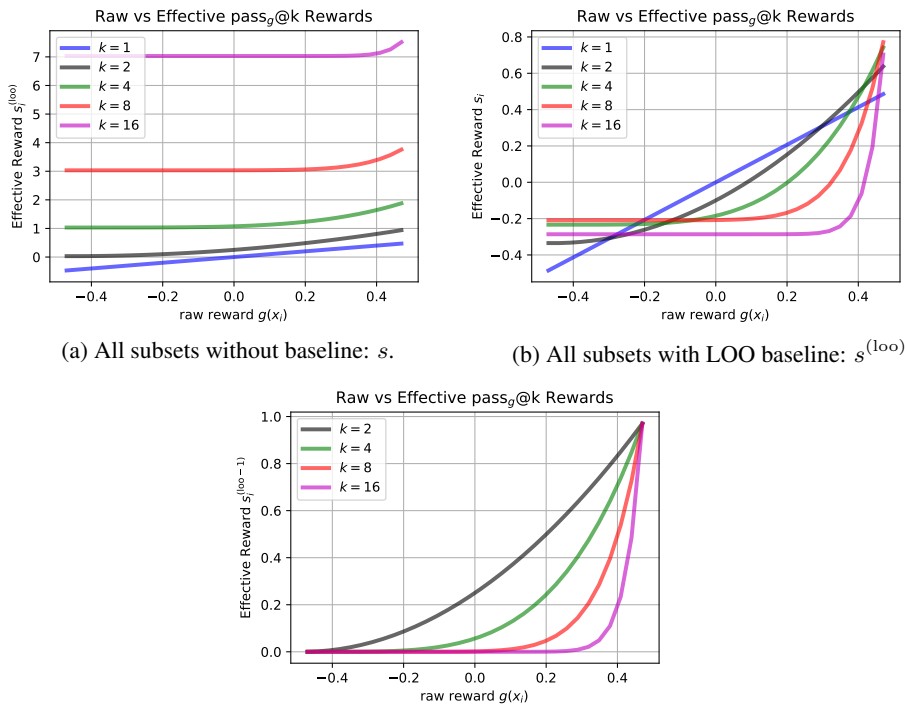

(a) All subsets without baseline: $s$.

(b) All subsets with LOO baseline: $s^{(\mathrm{loo})}$

(c) All $k$ sized subsets with $k-1$ sized subsets baseline: $s_i^{(\mathrm{loo}-1)}$

Figure 5: The effect of the LOO baseline on the effective rewards derived from $n = 32$ raw rewards $g(x_i)$ sampled uniformly from $[-1/2, +1/2]$. The non baselined effective rewards (a) from (19) include a vertical offset that grows with $k$ despite being a function of raw rewards (horizontal axis) that are centered around zero. The baselined effective rewards (b) and (c) from (29) and (33) respectively are more centered, and give rise to reduced gradient estimator variance. To construct the figure we grouped reward values into regularly spaced bins and averaged the transformed reward for each bin to construct the curves. *Note: because our transformations are from $\mathbb{R}^n \mapsto \mathbb{R}^n$ it is not possible to directly inspect a one-dimensional transformation.*

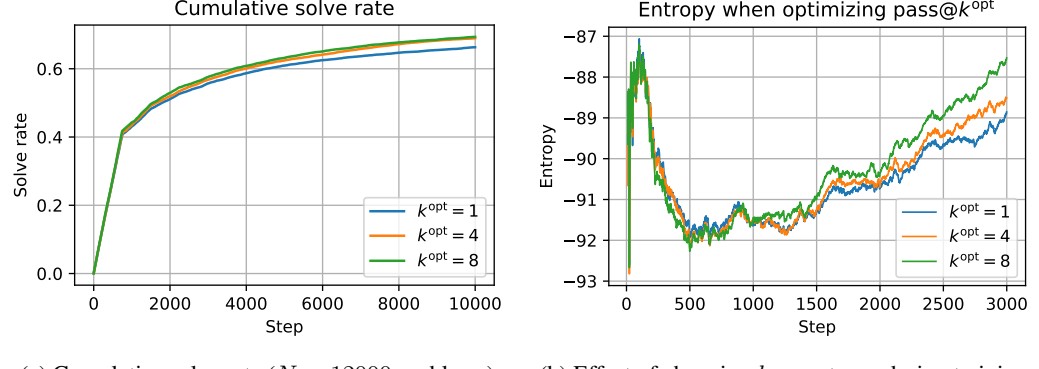

(a) Cumulative solve rate ($N = 12000$ problems)

(b) Effect of changing $k$ on entropy during training

Figure 6: (a): Increasing $k^{\mathrm{opt}}$ in PKPO training solves more problems during GEMMA2 RL. (b): A higher $k^{\mathrm{opt}}$ makes the model learn to have higher entropy during RL. Thus, by optimizing for pass@k with $k > 1$ instead of pass@1, the model tends to have higher entropy leading to better exploration and solving more problems. Note that the size of one epoch, which is 750 steps, is evident in (a), where we see the slope decrease at each epoch boundary.

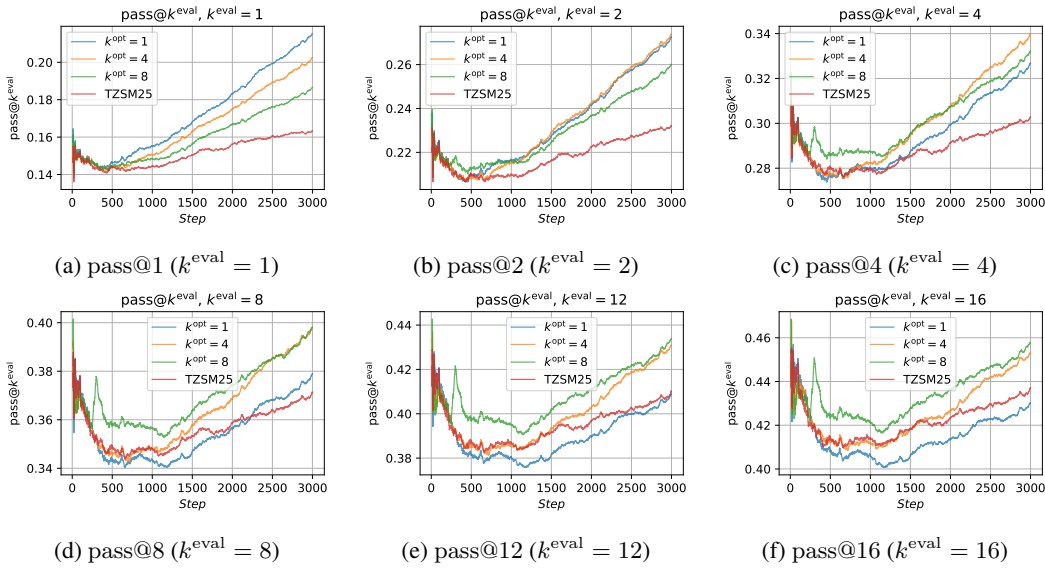

(a) pass@1 ($k^{\text{eval}} = 1$)  (b) pass@2 ($k^{\text{eval}} = 2$)  (c) pass@4 ($k^{\text{eval}} = 4$)

(d) pass@8 ($k^{\text{eval}} = 8$)  (e) pass@12 ($k^{\text{eval}} = 12$)  (f) pass@16 ($k^{\text{eval}} = 16$)

Figure 7: Effect of $k^{\text{opt}}$ (used in our PKPO training) on the rolling pass@$k^{\text{eval}}$ in GEMMA2 RL. Setting $k^{\text{opt}} = k^{\text{eval}}$ usually achieves the best pass@$k^{\text{eval}}$. Prior work [TZSM25] (which is equivalent to the specific case of $k^{\text{opt}} = n = 16$ in our notation) is also shown for comparison, and suffers here presumably due to the larger estimator variance and unreliable gradient (see also Figure 4).

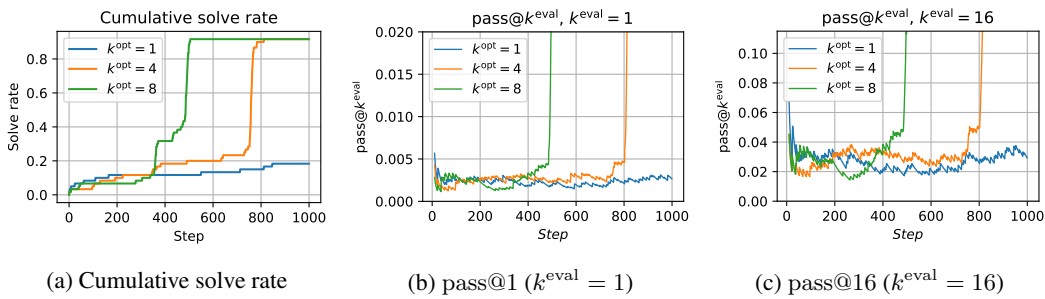

(a) Cumulative solve rate  (b) pass@1 ($k^{\text{eval}} = 1$)  (c) pass@16 ($k^{\text{eval}} = 16$)

Figure 8: Our PKPO ($k^{\text{opt}} > 1$) dramatically improves progress on the challenging `ARC-AGI-1`.

Table 7: Results for GEMMA2-2B on the `MATH` benchmark.

| GEMMA2-2B | k_eval=1 | k_eval=2 | k_eval=4 | k_eval=8 | k_eval=16 |
|---|---|---|---|---|---|
| k_opt=1 | **15.91** $\pm$ 0.40 | 18.12 $\pm$ 0.43 | 23.37 $\pm$ 0.48 | 29.58 $\pm$ 0.53 | 35.02 $\pm$ 0.58 |
| k_opt=2 | 15.15 $\pm$ 0.41 | **20.73** $\pm$ 0.45 | 25.81 $\pm$ 0.50 | 31.96 $\pm$ 0.55 | 37.75 $\pm$ 0.60 |
| k_opt=4 | 14.86 $\pm$ 0.43 | 19.86 $\pm$ 0.47 | **27.59** $\pm$ 0.52 | 34.27 $\pm$ 0.58 | 38.91 $\pm$ 0.62 |
| k_opt=8 | 14.19 $\pm$ 0.46 | 19.33 $\pm$ 0.50 | 26.45 $\pm$ 0.55 | **35.49** $\pm$ 0.60 | **40.73** $\pm$ 0.65 |
| [TZSM25] | 13.11 $\pm$ 0.50 | 18.09 $\pm$ 0.54 | 24.58 $\pm$ 0.60 | 31.81 $\pm$ 0.66 | 37.24 $\pm$ 0.71 |
| EntropyReg | 14.51 $\pm$ 0.48 | 18.95 $\pm$ 0.52 | 25.33 $\pm$ 0.58 | 30.95 $\pm$ 0.64 | 36.18 $\pm$ 0.69 |

Table 8: Results for GEMMA2-2B on the Coding benchmark.

| GEMMA2-2B | k_eval=1 | k_eval=2 | k_eval=4 | k_eval=8 | k_eval=16 |
|---|---|---|---|---|---|
| k_opt=1 | **19.82** $\pm$ 0.53 | 23.81 $\pm$ 0.57 | 29.75 $\pm$ 0.62 | 36.33 $\pm$ 0.66 | 42.04 $\pm$ 0.71 |
| k_opt=2 | 18.70 $\pm$ 0.54 | **26.94** $\pm$ 0.59 | 33.82 $\pm$ 0.64 | 40.95 $\pm$ 0.69 | 47.03 $\pm$ 0.74 |
| k_opt=4 | 18.69 $\pm$ 0.56 | 26.43 $\pm$ 0.61 | **36.81** $\pm$ 0.67 | 44.81 $\pm$ 0.73 | 50.55 $\pm$ 0.78 |
| k_opt=8 | 17.94 $\pm$ 0.59 | 25.86 $\pm$ 0.64 | 35.88 $\pm$ 0.70 | **46.45** $\pm$ 0.77 | **52.83** $\pm$ 0.83 |
| [TZSM25] | 16.81 $\pm$ 0.65 | 24.27 $\pm$ 0.69 | 33.11 $\pm$ 0.76 | 41.98 $\pm$ 0.84 | 47.26 $\pm$ 0.89 |
| EntropyReg | 18.05 $\pm$ 0.62 | 25.13 $\pm$ 0.67 | 34.01 $\pm$ 0.74 | 40.88 $\pm$ 0.81 | 46.15 $\pm$ 0.86 |

