# OpenReview forum: "Pass@K Policy Optimization: Solving Harder Reinforcement Learning Problems"
_NeurIPS.cc/2025/Conference — NeurIPS 2025 spotlight_

### Official Review · Reviewer_jhr4 · 2025-06-14

**Clarity:** 4
**Significance:** 3
**Originality:** 3
**Rating:** 5
**Confidence:** 4

**Summary:**

In this paper, "Solving Harder RL Tasks with Direct Pass@k Optimization", the authors introduce a specific transformation on the final rewards which leads to a direct optimization of the pass@k objective.

The primary contribution is the theoretical motivation for the proposed transformation. Specifically the authors introduce unbiased estimators for the pass@k objective and gradient in the case of binary rewards. These estimators are then extended to the case of real rewards. The authors further propose various baselines for variance reduction for both binary and real valued rewards, and show that these can be efficient computed.

Finally the authors show some empirical results of training with the transformed rewards on some toy and real domains.

**Questions:**

1. Compute restrictions aside were there any considerations for running $k^{eval}$ up to just 16? I would expect some sort of asymptotic behavior here and it would be interesting to observe this on, say, MATH
2. The ability to implement an annealing procedure for k<n feels like it has a lot of promise for practitioners. Extending these results might provide just the small-scale but robust empirical justification for this reward scaling suitable to a theoretical paper.
3. The step-function improvement on ARC-AGI-1 is pretty surprising (especially for the pass@1 eval). What's going on there?

**Ethical Concerns:**

["NO or VERY MINOR ethics concerns only"]

**Final Justification:**

I retain my rating of 5-Accept. The authors appear to have address most concerns. I would still like to see the title changed (as author's have sort of agreed to).

**Limitations:**

yes

**Quality:**

3

**Strengths And Weaknesses:**

Strengths:
1. The paper is clearly presented and easy to follow without extraneous content
2. The paper presents a timely and important contribution. The implicit misalignment between the pass@1 training and pass@k usage of models leaves much on the table. The paper's contribution, to provide a more general (and practical) reward transformation to support the direct pass@k, is, to my mind, a clearly valuable contribution with immediate application.

Weaknesses:
1. There is likely some missing literature. Specifically, there are results like (https://arxiv.org/abs/2502.07154) which would both help to further motivate the problem and better contextualize the current contribution.
2. The Introduction's positioning of this work as "...unblock(ing) the path to artificial super intelligence..." seems unnecessary
3. The paper title itself appears somewhat misleading. While the authors show _some_ results which, expectedly, demonstrate improvement on "harder" tasks, it is clear that the paper's primary contribution is a theoretical advancement. I would prefer to have seen the title more closely match the content.
4. ^^ Relatedly, the empirical validation, while suitable for a theoretically focused paper, is weak when making the more general claim suggested by the title (small models, few datasets).

---

> ### Author Response · Authors · 2025-07-31
> **Initial Rebuttal**
>
> Thanks for your time and effort, and supportive and critical comments. Please see shared comments above as well as those specific to you, here.
>
> > *There is likely some missing literature. Specifically, there are results like (https://arxiv.org/abs/2502.07154) which would both help to further motivate the problem and better contextualize the current contribution.*
>
> Thanks for the pointer to this interesting paper. We will add a reference to it in our own work that highlights the excellent motivating descriptive theory of their section 4.2. For the purposes of this rebuttal, we have some detailed comments on that paper that highlight key features of our own:
>
> - We really enjoyed their descriptive theory in section 4.2, which does indeed help to contextualize our work.
> - Generally, computing the `pass@1` (which they employ in their theory and hint at in their loss function) is intractable as it requires summing over all correct solutions.
> - Unlike our RL scheme, their objective of equation (3) does not optimize the `pass@k` (nor does it explicitly claim to), but is a heuristic that is suggested by the `1-(1-pass@1)^k` form of the `pass@k`.
> - From what we understand they seem to optimize their loss in a supervised learning setting given known ground truth solutions. In contrast, our approach finds correct solutions with RL while optimizing the `pass@k`. Their scheme may be powerful for distillation type schemes but it is not comparable to ours in its intention.
> - Their method boils down to a supervised learning loss function on a single sample, whereas ours boils down to a reinforcement learning reward transformation on a batch of n >= k samples. The use of n >= k samples is important; we can prove that no unbiased estimator of the `pass@k` exists that uses `n < k` samples. It is a neat proof that we are happy to provide here add to the appendix of our submission.
> - They compare with GRPO, which is not related to `pass@k` but rather is a basic variance reduction method for RL on `pass@1`. Our experiments try to show that our method works as intended by comparing to the most closely related methods that address the same problem, and by directly looking at the variance of the different schemes on a toy problem.
>
> [CTJ+21] Evaluating large language models trained on code, OpenAI.

---

> ### Author Response · Authors · 2025-08-05
> **rebuttals now visible**
>
> Thanks again for your time and effort. We made a mistake in OpenReview and our rebuttals were not visible to all. Our apologies - please see the rebuttals which should now be visible. We remain very confident in our theory as explained in our feedback to Reviewer Enpe.
>
> In case there is still an issue, here is an exact copy of the response to reviewer Enpe. Please let us know if any other responses are not visible.
>
> > (begin copy)
>
> Thanks for your time and effort, and supportive and critical comments. Please see shared comments above as well as those specific to you, here.
>
> Thank you for the detailed technical questions. We believe there may be a misunderstanding on these points, which we hope to clarify below. The unbiasedness of our estimators is a cornerstone of our work, and we appreciate the opportunity to elaborate on the proofs.
>
> ## Technical clarifications
>
> > *In Equation (8) ... [the result is incorrect] ... contradicting the claim of r_i being unbiased.*
>
> **We respectfully believe that the above statement in the review is incorrect.** See theorem 2 and the proof of that theorem. We assume that the reviewer missed this theorem given their subsequent comment *"Please provide a more detailed justification and proof for ... Equation (8)"*. We hope this clarification, along with the proof of Theorem 2, fully addresses the reviewer's concern.
>
> > *All theoretical proofs (e.g., Theorem 1, Theorem 2) assume k independent and identically distributed (i.i.d.) samples. However, in practical LLM fine-tuning settings, samples from the same mini-batch often share components like dropout masks and KV-cache. This makes them non-independent, directly violating a core assumption of the theoretical guarantees and potentially invalidating the unbiasedness claims in the empirical context.*
>
> **We respectfully believe that this statement is incorrect:** Conditional independence is guaranteed by the standard sampling process of autoregressive models, it is not an assumption we are making. Given a prompt, each sample is generated independently. Neither the use of a shared KV-cache for the prompt encoding nor standard dropout violates this.
>
> > *The paper claims that `s^{(loo-1)}` is an unbiased estimator despite admitting a biased baseline ... undermining the claim of an unbiased gradient estimator.*
>
> **We respectfully believe this is incorrect.** The unbiasedness here is a subtle point and we are glad for the chance to expound on it. It is a cornerstone of our work and also [TZSM25] where it is Lemma 1; if the reviewer comment were correct, that paper would also be deeply flawed (which it is not). The unbiasedness also follows from Corollary 1 in our submission, but some extra work given that the baseline is not constant but rather depends on other samples. In detail:
>
> Assume w.l.o.g. we have two samples `x_1` and `x_2` and an arbitrary function `g(x_2)` as the baseline for the estimator for the gradient of the reward `r(x_1)`.
>
> The baselined policy gradient estimator is:
>
> `∇_θ J ≈ (r(x_1) - g(x_2)) * ∇_θ log p(x_1 | z, θ)`
>
> Consider the expectation over prompts `z` from the dataset `D` and conditionally independent samples `x_1`, `x_2` from the policy `p(·|z, θ)`.
>
> `E[g(x_2) * ∇_θ log p(x_1 | z, θ)]`
>
> Using iterated expectations, we can first condition on a fixed prompt `z`:
>
> `= E_{z∼D} [ E_{x_1, x_2∼p(·|z,θ)} [ g(x_2) * ∇_θ log p(x_1 | z, θ) | z ] ]`
>
> The inner expectation of the product can be separated:
>
> `= E_{z∼D} [ E_{x_2∼p(·|z,θ)}[g(x_2)] * E_{x_1∼p(·|z,θ)}[∇_θ log p(x_1 | z, θ)] ]`
>
> The second term is always zero by Lemma 2 from our submission, i.e.:
>
> `E_{x_1∼p(·|z,θ)}[∇_θ log p(x_1 | z, θ)] = ∫ p(x_1|z,θ) * (∇_θ p(x_1|z,θ) / p(x_1|z,θ)) dx_1 = ∇_θ ∫ p(x_1|z,θ) dx_1 = ∇_θ(1) = 0`
>
> so the inner expectation becomes zero for any prompt `z`:
>
> `= E_{z∼D} [ E_{x_2∼p(·|z,θ)}[g(x_2)] * 0 ] = E_{z∼D}[0] = 0`
>
> The introduced bias is therefore equal to zero.
>
> ## Other responses
>
> > the provided Python implementation (Listing 1) explicitly notes that it costs `O(nk+n log(n))`
>  for simplicity ... provide a clear path towards achieving an implementation with implementation with complexity as claimed in Theorem 5
>
> Please see lines 414-419 of the supplementary material.
>
> > O(n log n) ... significantly impacts scalability ...
>
> As per lines 72-79, n is the number of samples for a given task (such as a single math question) in a single minibatch. As such this cost was negligible compared to the gradient and sampling costs in our experiments (which for a typical quadratic attention transformer costs `O(f(|theta|) n t^2)` where `t` is the sequence length and `f(|theta|)` is a function of the number of parameters). We were surprised that such an efficient algorithm exists given that the naive cost is `O(n choose k)`, and we hope the reviewer appreciates the effort it took to find these efficient algorithms.
>
> > (end copy)

---

> > ### Author Response · Authors · 2025-08-05
> > **Details of experiments**
> >
> > > Compute restrictions aside were there any considerations for running up to just 16? I would expect some sort of asymptotic behavior here and it would be interesting to observe this on, say, MATH
> >
> > No, the decision was largely based on compute restrictions. The choice of `n=16` was the sweet spot of having a large enough sweep over k while avoiding the memory management challenges with large batch sizes. We agree however, that the asymptotic behaviour of increasing k would be interesting to investigate, especially for applications like alphaproof which require generating a large number of diverse candidates
> >
> > > The ability to implement an annealing procedure for k<n feels like it has a lot of promise for practitioners. Extending these results might provide just the small-scale but robust empirical justification for this reward scaling suitable to a theoretical paper.
> >
> > We have added a thorough set of new experiments which should address most of the empirical concerns raised by all reviewers. Please refer to the common comments for more details. While we did not have the bandwidth to run more annealing experiments in the time-frame of the rebuttal, our sample experiment in the paper combined with the strong trend of `k_opt` matching `pass@k_eval` shown in our newer experiments should be sufficient for a strong empirical justification
> >
> >
> > > The step-function improvement on ARC-AGI-1 is pretty surprising (especially for the pass@1 eval). What's going on there?
> >
> > The sudden climb in Figure 5 corresponds to the model learning to solve a big chunk of the task set within a range of a few steps. Note that the cumulative solve rate also jumps to ~90% around the same mark. Based on our manual inspection, it seems the task set has multiple similar problems ultimately splitting the set into a few different difficulty levels (even within the easy subset). The multiple distinct step improvements until convergence seem to be the model learning to solve multiple problems of a certain difficulty level at the same time (with the last jump being the largest one)

---

> > > ### Comment · Reviewer_jhr4 · 2025-08-08
> > > **Reply to rebuttal**
> > >
> > > Thanks for the response and engagement. I appreciate some of the clarifications.
> > >
> > > I do not mean to suggest any action on your part with this comment, but if indeed there are "levels" of shared difficulty contained within ARC-1 (from the perspective of the model), it would be an interesting and useful community contribution.

---

### Official Review · Reviewer_vPUj · 2025-07-02

**Clarity:** 3
**Significance:** 2
**Originality:** 2
**Rating:** 4
**Confidence:** 2

**Summary:**

This paper focuses on optimising policies jointly over multiple samples, rather than following the usual single-sample optimisation paradigm. Anticipating the fact that multiple attempts will be made at inference time, it enables training policies that have a higher chance of producing at least one very good output. The paper proposes optimising over the best k out of n samples, thereby generalising prior work which fixes k = n. Additionally, it introduces an unbiased and low-variance gradient estimator for this setting, supported by mathematical proofs, and provides experiments to evaluate its effect on performance during training.

**Questions:**

What is the statistical significance of the empirical results? Specifically, how many seeds were used per experiment?

Can we reasonably expect this method to outperform a standard entropy-regularized baseline? It would be helpful to understand whether the benefits go beyond what entropy regularisation alone can achieve.

**Ethical Concerns:**

["NO or VERY MINOR ethics concerns only"]

**Final Justification:**

I am willing to support the paper, although with a low confidence.

The paper seems to make important theoretical contributions. And support them with empirical evidence. The initial experiments lack statistical reports, but from the rebuttal, I trust that the authors will address this in the revised manuscript. I also trust that the authors will update their title and reduce the breadth of their claim to their mathematical contribution.

I regret that the others reviewers did not engage more in the discussions, which would have helped the assessment of this paper. From the information I have, I am leaning towards a weak accept.

**Limitations:**

There does not seem to be any discussion of the limitations. Could the authors add one?

**Quality:**

3

**Strengths And Weaknesses:**

**Strengths:**
- The pass@k optimization paradigm is relevant and increasingly important in RL for generative models. This paper contributes to it by introducing a general formulation (arbitrary k≤n), supporting it with rigorous mathematical derivations, and providing empirical validation.
- The paper is well written. The mathematical content is clearly presented, and the proofs are systematically included, either in the main text or in the appendix.
- The experimental section includes a toy problem to validate and illustrate the key claims (i.e., improved sample performance and reduced gradient variance).
- The paper also presents large-scale experiments using Gemma-2 on challenging benchmarks (e.g., MATH, ARC-AGI), demonstrating that the proposed method can be applied to realistic and difficult problems.



**Weaknesses:**
- The title feels too broad relative to the empirical scope of the paper. While it refers to “solving harder RL tasks,” the experiments are limited to two tasks in the LLM domain. The claim would be more accurate if it were restricted to LLMs.
- There already exist methods designed to jointly optimize over multiple attempts in hard RL tasks, such as Grinsztajn et al. (2022), Chalumeau et al. (2023), and follow-ups like Hottung et al. (2024) and Bonnet et al. (2024). It is unclear whether the approach proposed here would outperform these methods in other domains (e.g., combinatorial optimization). To be fully fair, the claims should be qualified to the LLM context, where those baselines have not (to my knowledge) been evaluated.
- Statistical rigor: It is unclear whether the experimental results are averaged over multiple seeds. If only a single seed is used throughout, this raises concerns about the statistical robustness of the findings.
- Line 54/55 claims that “[CTV19] and [XDC+20] present elegant approximations that are rather general but less efficient in our setting.” However, no empirical or theoretical justification is provided for this efficiency claim.
- Baselines (concurrent methods): I am surprised by the lack of baselines in the paper. Is there only one concurrent method to compare to?
- I’m unclear on what the loss function contributes beyond encouraging high-entropy solutions. - It would be useful to compare against a baseline that optimizes pass@1 (or pass@n) with an added entropy regularization term. This would help disentangle the benefits of the proposed method from standard exploration techniques in RL, such as entropy bonuses, which are widely used and easy to implement.
- In Section 5.2.3, the authors demonstrate promising results on a difficult benchmark, but do not compare against other methods. The results would be compelling if competing approaches were also shown to fail in this setting, similar to the k=1 baseline. It remains unclear how well [TZSM25] or other baselines would perform on this task.

References:
- Nathan Grinsztajn, Daniel Furelos-Blanco, Shikha Surana, Clément Bonnet, and Thomas D. Barrett. Winner takes it all: Training performant rl populations for combinatorial optimization. Neurips 2023.
- Felix Chalumeau, Shikha Surana, Clément Bonnet, Nathan Grinsztajn, Arnu Pretorius, Alexandre Laterre, and Thomas D Barrett. Combinatorial optimization with policy adaptation using latent space search. Neurips 2023
- Andre Hottung, Mridul Mahajan, and Kevin Tierney. Polynet: Learning diverse solution strategies for neural combinatorial optimization. ICLR 2024
- Clement Bonnet, Matthew Macfarlane. Searching Latent Program Spaces.

---

> ### Author Response · Authors · 2025-07-31
> **Initial Rebuttal**
>
> Thanks for your time and effort, and supportive and critical comments. Please see shared comments above as well as those specific to you, here.
>
> > Statistical rigor ... if only a single seed is used throughout, this raises concerns about the statistical robustness of the findings.
>
> This is a fair and important point. Due to time and computational constraints, we were unable to run all experiments over multiple seeds for this rebuttal. We acknowledge this as a limitation and will add a note to the paper. We will make it a priority to add results with multiple seeds to the camera-ready version.
>
> > *"[CTV19] and [XDC+20] present elegant approximations that are rather general but less efficient in our setting." However, no empirical or theoretical justification is provided for this efficiency claim.*
>
> We consider all subsets of size `k` of `1,2,...,n`. Naively applying their generic methods would have cost at least `O(n choose k)`.
>
> > Baselines (concurrent methods): I am surprised by the lack of baselines in the paper. Is there only one concurrent method to compare to?
>
> Yes! There is only one (very recent) method that derives unbiased estimators for the `pass@k`, [TZSM25]. However it only considers number of samples `n=k`. Or work goes much further in theoretical depth and is more generally applicable (by decoupling the minibatch size and `k`). However, their insight of using subsets of size `k-1` in the baseline was new to us and dramatically improved our estimators, leading to our `s^{loo-1}` (which was also by far the most challenging to derive in our setup, but to our suprise can be computed efficiently).
>
> > In Section 5.2.3, the authors demonstrate promising results on a difficult benchmark, but do not compare against other methods. The results would be compelling if competing approaches were also shown to fail in this setting, similar to the k=1 baseline. It remains unclear how well [TZSM25] or other baselines would perform on this task.
>
> We have run this for [TZSM25] and it indeed performs poorly. We will include this plot in our paper as it is indeed good evidence indicative of the strength of our method, thanks for the suggestion.
>
> > There already exist methods designed to jointly optimize over multiple attempts in hard RL tasks, such as Grinsztajn et al. (2022), Chalumeau et al. (2023), and follow-ups like Hottung et al. (2024) and Bonnet et al. (2024). ... To be fully fair, the claims should be qualified to the LLM context, where those baselines have not (to my knowledge) been evaluated.
>
> Indeed, these are very interesting but very different to our drop in reward transformations for RL that guarantee `pass@k` optimization. We agree that lots of great work could be done combining these with RL and LLMs.

---

> > ### Comment · Reviewer_vPUj · 2025-08-05
> >
> > I thank the authors for their additional response and for sharing new empirical evidence. I have read these additions as well as the comments from other reviewers.
> >
> > Parts of my original concerns have been addressed, and I trust the authors will implement some remaining updates (e.g. changing the title, running experiments with multiple seeds, adding error bars, and including an appropriate baseline) to resolve any outstanding experimental points.
> >
> > Other reviewers have raised equally valid points, and I am not certain they have all been addressed. In particular, Reviewer Enpe has highlighted potential weaknesses in the theoretical framework that do not appear to have been discussed in the authors’ rebuttal.
> >
> > I could agree to support the paper’s acceptance despite some experimental limitations if there were full consensus on the theoretical issues, but that does not yet seem to be the case.
> >
> > Could the authors please address or at least comment on the theoretical concerns raised by Reviewer Enpe?

---

> > > ### Author Response · Authors · 2025-08-05
> > > **rebuttals were not visible - our apologies**
> > >
> > > Thanks for raising this and thanks again for your time and effort. We made a mistake in OpenReview and our rebuttals were not visible to all. Our apologies - please see the rebuttals which should now be visible. We remain very confident in our theory as explained in our feedback to Reviewer Enpe.
> > >
> > > In case there is still an issue, here is an exact copy of the response to reviewer Enpe:
> > >
> > > > (begin copy)
> > >
> > > Thanks for your time and effort, and supportive and critical comments. Please see shared comments above as well as those specific to you, here.
> > >
> > > Thank you for the detailed technical questions. We believe there may be a misunderstanding on these points, which we hope to clarify below. The unbiasedness of our estimators is a cornerstone of our work, and we appreciate the opportunity to elaborate on the proofs.
> > >
> > > ## Technical clarifications
> > >
> > > > *In Equation (8) ... [the result is incorrect] ... contradicting the claim of r_i being unbiased.*
> > >
> > > **We respectfully believe that the above statement in the review is incorrect.** See theorem 2 and the proof of that theorem. We assume that the reviewer missed this theorem given their subsequent comment *"Please provide a more detailed justification and proof for ... Equation (8)"*. We hope this clarification, along with the proof of Theorem 2, fully addresses the reviewer's concern.
> > >
> > > > *All theoretical proofs (e.g., Theorem 1, Theorem 2) assume k independent and identically distributed (i.i.d.) samples. However, in practical LLM fine-tuning settings, samples from the same mini-batch often share components like dropout masks and KV-cache. This makes them non-independent, directly violating a core assumption of the theoretical guarantees and potentially invalidating the unbiasedness claims in the empirical context.*
> > >
> > > **We respectfully believe that this statement is incorrect:** Conditional independence is guaranteed by the standard sampling process of autoregressive models, it is not an assumption we are making. Given a prompt, each sample is generated independently. Neither the use of a shared KV-cache for the prompt encoding nor standard dropout violates this.
> > >
> > > > *The paper claims that `s^{(loo-1)}` is an unbiased estimator despite admitting a biased baseline ... undermining the claim of an unbiased gradient estimator.*
> > >
> > > **We respectfully believe this is incorrect.** The unbiasedness here is a subtle point and we are glad for the chance to expound on it. It is a cornerstone of our work and also [TZSM25] where it is Lemma 1; if the reviewer comment were correct, that paper would also be deeply flawed (which it is not). The unbiasedness also follows from Corollary 1 in our submission, but some extra work given that the baseline is not constant but rather depends on other samples. In detail:
> > >
> > > Assume w.l.o.g. we have two samples `x_1` and `x_2` and an arbitrary function `g(x_2)` as the baseline for the estimator for the gradient of the reward `r(x_1)`.
> > >
> > > The baselined policy gradient estimator is:
> > >
> > > `∇_θ J ≈ (r(x_1) - g(x_2)) * ∇_θ log p(x_1 | z, θ)`
> > >
> > > Consider the expectation over prompts `z` from the dataset `D` and conditionally independent samples `x_1`, `x_2` from the policy `p(·|z, θ)`.
> > >
> > > `E[g(x_2) * ∇_θ log p(x_1 | z, θ)]`
> > >
> > > Using iterated expectations, we can first condition on a fixed prompt `z`:
> > >
> > > `= E_{z∼D} [ E_{x_1, x_2∼p(·|z,θ)} [ g(x_2) * ∇_θ log p(x_1 | z, θ) | z ] ]`
> > >
> > > The inner expectation of the product can be separated:
> > >
> > > `= E_{z∼D} [ E_{x_2∼p(·|z,θ)}[g(x_2)] * E_{x_1∼p(·|z,θ)}[∇_θ log p(x_1 | z, θ)] ]`
> > >
> > > The second term is always zero by Lemma 2 from our submission, i.e.:
> > >
> > > `E_{x_1∼p(·|z,θ)}[∇_θ log p(x_1 | z, θ)] = ∫ p(x_1|z,θ) * (∇_θ p(x_1|z,θ) / p(x_1|z,θ)) dx_1 = ∇_θ ∫ p(x_1|z,θ) dx_1 = ∇_θ(1) = 0`
> > >
> > > so the inner expectation becomes zero for any prompt `z`:
> > >
> > > `= E_{z∼D} [ E_{x_2∼p(·|z,θ)}[g(x_2)] * 0 ] = E_{z∼D}[0] = 0`
> > >
> > > The introduced bias is therefore equal to zero.
> > >
> > > ## Other responses
> > >
> > > > the provided Python implementation (Listing 1) explicitly notes that it costs `O(nk+n log(n))`
> > >  for simplicity ... provide a clear path towards achieving an implementation with implementation with complexity as claimed in Theorem 5
> > >
> > > Please see lines 414-419 of the supplementary material.
> > >
> > > > O(n log n) ... significantly impacts scalability ...
> > >
> > > As per lines 72-79, n is the number of samples for a given task (such as a single math question) in a single minibatch. As such this cost was negligible compared to the gradient and sampling costs in our experiments (which for a typical quadratic attention transformer costs `O(f(|theta|) n t^2)` where `t` is the sequence length and `f(|theta|)` is a function of the number of parameters). We were surprised that such an efficient algorithm exists given that the naive cost is `O(n choose k)`, and we hope the reviewer appreciates the effort it took to find these efficient algorithms.
> > >
> > > > (end copy)

---

> > > > ### Author Response · Authors · 2025-08-05
> > > > **Addressing empirical concerns and details of experiments**
> > > >
> > > > We have added a thorough set of new experiments which should address most of the empirical concerns raised by all reviewers. Please refer to the common comments for more details. Experiements Part 1/3 includes details of the experiments, and Parts 2 and 3 include detailed tables
> > > >
> > > > >Statistical rigor: It is unclear whether the experimental results are averaged over multiple seeds. If only a single seed is used throughout, this raises concerns about the statistical robustness of the findings.
> > > >
> > > > We show statistical significance by running the new experiment with 3 seeds. We show the gains hold on held out test sets with standard error reported and are statistically significant
> > > >
> > > > > **Baselines**: compare against a baseline that ... added entropy regularization term. This would help disentangle the benefits of the proposed method from standard exploration techniques in RL
> > > >
> > > > We run new experiments with both TZSM25 and the entropy regularization baseline. We tune the latter by running a small sweep on the `entropy_coefficient`. Our method out performs both these baselines in all 3 benchmarks and all 3 models that we evaluate on. Please refer to common comments for more details
> > > >
> > > > >  It remains unclear how well [TZSM25] or other baselines would perform on the difficult benchmark (ARC-AGI)
> > > >
> > > > Please refer to the new experiments part 3/3. Both the baselines (TZSM and Entropy regularization) are sub-optimal. Only our method with high `k_opt=4,8` explores well enough and unblocks learning

---

> ### Comment · Reviewer_vPUj · 2025-08-06
>
> Thanks for your reply, and for sharing these additional elements. I am curious to hear the other reviewers' opinion to your rebuttal. If the other reviewers do not submit their updated recommendations, I will nevertheless raise my score to a weak accept (4), trusting that the authors will address the points discussed.

---

### Official Review · Reviewer_Enpe · 2025-07-05

**Clarity:** 2
**Significance:** 2
**Originality:** 2
**Rating:** 4
**Confidence:** 1

**Summary:**

This paper introduces Pass-at-k Policy Optimization (PKPO), a method for optimizing the pass@k metric in Reinforcement Learning rather than the conventional pass@1 approach. The key insight is that when sampling multiple solutions per problem, optimizing for the probability that at least one of k samples succeeds can improve exploration and performance on challenging tasks. The authors provide mathematical foundations, including unbiased estimators for pass@k and its gradient in both binary and continuous reward settings. The method is validated on toy problems and a small subset of real-world experiments using Gemma-2 on mathematical reasoning tasks.

**Questions:**

1. Given that practical RL fine-tuning often involves non-i.i.d. samples (e.g., due to shared dropout, KV-cache), how do these violations of the i.i.d. assumption affect the theoretical guarantees of unbiasedness and variance reduction? Can the authors provide theoretical analysis or empirical validation under these realistic, non-i.i.d. conditions to demonstrate the robustness of PKPO?
2.   The theoretical claim of an unbiased gradient estimator is crucial. Can the authors provide a rigorous proof that the overall gradient estimator remains unbiased when using $s^{(loo-1)}$, specifically addressing how the potential admitted bias in the baseline term within the score-function product does not propagate to make the entire gradient estimate biased?
3. * Can the authors provide a clear path towards achieving an implementation with $O(k + n \log n)$ complexity as claimed in Theorem 5, rather than the currently noted $O(nk + n \log n)$?
    * How do you propose to address the super-linear scaling in wall-clock time across multiple tasks that require sorting, particularly for large-scale applications with many tasks?
4.  Please provide a more detailed justification and proof for the specific form of $r_i$ in Equation (8), especially for the $f_i=0$ case and its interaction with the count `c`. How does this formulation ensure unbiasedness for all $k \le n$, and specifically, address concerns about potential bias for small $k$ values?
5. To strengthen the theoretical contributions and their applicability, can the authors provide experiments on domains or tasks where the i.i.d. assumption (or approximations thereof) is better satisfied or can be explicitly controlled? This would help validate the method's performance under its theoretical premises.

**Ethical Concerns:**

["NO or VERY MINOR ethics concerns only"]

**Final Justification:**

I am not an expert in the field. I am trying to evaluate this paper based on the mathematical rigor and empirical results.

**Limitations:**

The authors acknowledge some limitations but miss several critical issues. These unaddressed points fundamentally impact the method's theoretical soundness and practical applicability. Specifically, the paper does not adequately discuss: (1) the potential bias in the leave-one-out formulation for $r_i$ in Equation (8), (2) the violation of i.i.d. assumptions in practical RL settings for LLMs, (3) the computational scaling issues across multiple tasks due to sorting, and (4) the potential for gradient bias originating from the admitted bias in the baseline terms within score-function estimators. Addressing these fundamental problems is crucial for the work.

**Quality:**

2

**Strengths And Weaknesses:**

This paper addresses an important problem in reinforcement learning: how to effectively optimize when multiple solution attempts per problem are sampled. The core idea of directly optimizing for pass@k is valuable. However, the proposed method suffers from several fundamental methodological and computational issues that undermine its theoretical claims and practical scalability.

**Strengths:**
*  The core insight of directly optimizing pass@k rather than pass@1 is valuable, as it addresses a real limitation in current RL practice where multiple samples are commonly used but optimization focuses only on individual sample quality[cite: 2, 4].
* The approach of deriving unbiased estimators for pass@k and its gradient, applicable to both binary and continuous reward settings, is sophisticated and aims to provide a complete theoretical treatment.

**Weaknesses:**
*  The paper contains several mathematical flaws that compromise its theoretical rigor:
    *  In Equation (8), the `r_i` definition for $f_i=0$ (incorrect samples) uses $\rho(n-1,c,k-1)$. This formulation, particularly how the count `c` is used for samples that are assumed incorrect, appears to include the incorrect sample being evaluated, which violates the leave-one-out principle. This can introduce bias, especially for small `k`, directly contradicting the claim of `r_i` being unbiased.
    * All theoretical proofs (e.g., Theorem 1, Theorem 2) assume `k` independent and identically distributed (i.i.d.) samples. However, in practical LLM fine-tuning settings, samples from the same mini-batch often share components like dropout masks and KV-cache. This makes them non-independent, directly violating a core assumption of the theoretical guarantees and potentially invalidating the unbiasedness claims in the empirical context.
    * The paper claims that $s^{(loo-1)}$ is an unbiased estimator despite admitting a biased baseline. However, when a biased baseline appears directly inside the score-function product (as implied by the general form of policy gradient estimators), the overall gradient estimate itself becomes biased. This specific technical issue is not adequately addressed, undermining the claim of an unbiased gradient estimator.
* **some computational complexity issues**
    * Theorem 5 claims an $O(k + n \log n)$ complexity for computing the reward transformations. However, the provided Python implementation (Listing 1) explicitly notes that it "costs $O(nk + n \log n)$ for simplicity". This discrepancy is significant for scalability.
    * For continuous rewards, the method requires $O(n \log n)$ sorting per task (due to the assumption that rewards $g_i$ are sorted in Theorem 3 and its application to `s_i` computation), which would make the wall-clock time super-linear in `n` across multiple tasks. This significantly impacts scalability for large `k` or when processing many tasks.
* The experimental evaluation is limited to one specific model architecture (Gemma-2 2B variant) and primarily two domains (mathematical reasoning and logic puzzles). This narrow scope limits the empirical contribution and does not validate the method's effectiveness on diverse RL applications or in settings where the theoretical assumptions might hold better.
* The comparison largely focuses on vanilla pass@1 optimization. The paper lacks comprehensive comparisons with other contemporary inference-time optimization methods, advanced exploration techniques, or alternative approaches for handling multiple samples in RL literature.

---

> ### Author Response · Authors · 2025-07-31
> **Initial Rebuttal**
>
> Thanks for your time and effort, and supportive and critical comments. Please see shared comments above as well as those specific to you, here.
>
> Thank you for the detailed technical questions. We believe there may be a misunderstanding on these points, which we hope to clarify below. The unbiasedness of our estimators is a cornerstone of our work, and we appreciate the opportunity to elaborate on the proofs.
>
> ## Technical clarifications
>
> > *In Equation (8) ... [the result is incorrect] ... contradicting the claim of r_i being unbiased.*
>
> **We respectfully believe that the above statement in the review is incorrect.** See theorem 2 and the proof of that theorem. We assume that the reviewer missed this theorem given their subsequent comment *"Please provide a more detailed justification and proof for ... Equation (8)"*. We hope this clarification, along with the proof of Theorem 2, fully addresses the reviewer's concern.
>
> > *All theoretical proofs (e.g., Theorem 1, Theorem 2) assume k independent and identically distributed (i.i.d.) samples. However, in practical LLM fine-tuning settings, samples from the same mini-batch often share components like dropout masks and KV-cache. This makes them non-independent, directly violating a core assumption of the theoretical guarantees and potentially invalidating the unbiasedness claims in the empirical context.*
>
> **We respectfully believe that this statement is incorrect:** Conditional independence is guaranteed by the standard sampling process of autoregressive models, it is not an assumption we are making. Given a prompt, each sample is generated independently. Neither the use of a shared KV-cache for the prompt encoding nor standard dropout violates this.
>
> > *The paper claims that `s^{(loo-1)}` is an unbiased estimator despite admitting a biased baseline ... undermining the claim of an unbiased gradient estimator.*
>
> **We respectfully believe this is incorrect.** The unbiasedness here is a subtle point and we are glad for the chance to expound on it. It is a cornerstone of our work and also [TZSM25] where it is Lemma 1; if the reviewer comment were correct, that paper would also be deeply flawed (which it is not). The unbiasedness also follows from Corollary 1 in our submission, but some extra work given that the baseline is not constant but rather depends on other samples. In detail:
>
> Assume w.l.o.g. we have two samples `x_1` and `x_2` and an arbitrary function `g(x_2)` as the baseline for the estimator for the gradient of the reward `r(x_1)`.
>
> The baselined policy gradient estimator is:
>
> `∇_θ J ≈ (r(x_1) - g(x_2)) * ∇_θ log p(x_1 | z, θ)`
>
> Consider the expectation over prompts `z` from the dataset `D` and conditionally independent samples `x_1`, `x_2` from the policy `p(·|z, θ)`.
>
> `E[g(x_2) * ∇_θ log p(x_1 | z, θ)]`
>
> Using iterated expectations, we can first condition on a fixed prompt `z`:
>
> `= E_{z∼D} [ E_{x_1, x_2∼p(·|z,θ)} [ g(x_2) * ∇_θ log p(x_1 | z, θ) | z ] ]`
>
> The inner expectation of the product can be separated:
>
> `= E_{z∼D} [ E_{x_2∼p(·|z,θ)}[g(x_2)] * E_{x_1∼p(·|z,θ)}[∇_θ log p(x_1 | z, θ)] ]`
>
> The second term is always zero by Lemma 2 from our submission, i.e.:
>
> `E_{x_1∼p(·|z,θ)}[∇_θ log p(x_1 | z, θ)] = ∫ p(x_1|z,θ) * (∇_θ p(x_1|z,θ) / p(x_1|z,θ)) dx_1 = ∇_θ ∫ p(x_1|z,θ) dx_1 = ∇_θ(1) = 0`
>
> so the inner expectation becomes zero for any prompt `z`:
>
> `= E_{z∼D} [ E_{x_2∼p(·|z,θ)}[g(x_2)] * 0 ] = E_{z∼D}[0] = 0`
>
> The introduced bias is therefore equal to zero.
>
> ## Other responses
>
> > the provided Python implementation (Listing 1) explicitly notes that it costs `O(nk+n log(n))`
>  for simplicity ... provide a clear path towards achieving an implementation with implementation with complexity as claimed in Theorem 5
>
> Please see lines 414-419 of the supplementary material.
>
> > O(n log n) ... significantly impacts scalability ...
>
> As per lines 72-79, n is the number of samples for a given task (such as a single math question) in a single minibatch. As such this cost was negligible compared to the gradient and sampling costs in our experiments (which for a typical quadratic attention transformer costs `O(f(|theta|) n t^2)` where `t` is the sequence length and `f(|theta|)` is a function of the number of parameters). We were surprised that such an efficient algorithm exists given that the naive cost is `O(n choose k)`, and we hope the reviewer appreciates the effort it took to find these efficient algorithms.

---

> > ### Comment · Reviewer_vPUj · 2025-08-06
> >
> > I am really curious to hear Reviewer Enpe about this rebuttal. Would Reviewer Enpe agree that the theory is actually correct?

---

> > > ### Comment · Reviewer_Enpe · 2025-08-08
> > > **tentative accept**
> > >
> > > It took me some time to understand the theory and to read the comments of other reviewers. I tend to accept this paper, but with very low confidence.

---

> ### Author Response · Authors · 2025-08-05
> **questions?**
>
> Thanks again for your time and effort, and for diving deeply into the paper. We hope that your theoretical concerns have been addressed in our rebuttal. If not, we are happy to explain further, just let us know.

---

### Official Review · Reviewer_uuR2 · 2025-07-08

**Clarity:** 3
**Significance:** 3
**Originality:** 3
**Rating:** 5
**Confidence:** 4

**Summary:**

The authors derive a set of new algorithms for direct pass@k optimization. Starting from the unbiased pass@k /max@k estimators in the case of binary / continuous rewards, the authors derive a gradient estimator (both without and with variance reduction via LOO baseline).

The empirical evaluation involves training Gemma-2-2B on MATH and ARC-AGI-1 (easy subset), and baseline against another pass@k estimator (Tang et al, 2025). These experiments show improved pass@k performance when directly optimizing for the same k, as well as improvement over pass@k estimator from (Tang et al, 2025).

The authors also propose a method in which k is annealed during training, leading to models that balance better pass@1 and pass@k performance.

**Questions:**

1. Can you give an intuitive explanation for the resulting adjustments to the gradient estimator? (Equations 8 and 18-21)

2. Do you have evidence that optimizing for pass@k leads to higher diversity of solutions beyond simple metrics like entropy?

**Ethical Concerns:**

["NO or VERY MINOR ethics concerns only"]

**Final Justification:**

I increased the rating as the authors addressed my main concerns about the lack of empirical support (points 1-5 on the list of Weaknesses).

I've been also convinced that BoN approximation baseline makes less sense (point 5) and the cumulative solve rate is a good indication of diversity here (point 7). However, I'd still argue for the inclusion of the GRPO baseline (point 5) and the inclusion of semantic measures of diversity (point 6) (see discussion for more details).

**Limitations:**

None.

**Paper Formatting Concerns:**

Already highlighted the issue with formatting citations.

**Quality:**

3

**Strengths And Weaknesses:**

**Summary:** the theoretical contribution of this work is clearly strong; the main crux is whether the much weaker empirical section is sufficient to justify acceptance. In this reviewer's opinion, this paper does not meet the bar for the strength of empirical support in RL-for-LMs theoretical papers published at NeurIPS, hence Borderline Reject. However, I'd happily change my rating to Accept with the addition of several new experiments (see the Weaknesses section).


## Strengths

The paper is very well-written and addresses an important problem: improving the effectiveness of multiple trials by directly optimizing for pass@k / max@k. The derived estimators are (to my knowledge) all novel contributions, and provide a clear improvement over the prior work by generalizing to continuous rewards and cases where sample size $n \neq k$, as well as reducing the variance of the estimator. I didn't find any issues with the proofs, although it is possible I missed some.

The method is also very straightforward to apply, since it just transforms the rewards (python implementation in Listing 1 is a nice addition).

The demonstrated results are quite promising: pass@k performance is generally better when optimizing for the same (or the at least closest) k and consistently outperforms another estimator (Tang et al, 2025). The results are particularly promising for ARC-AGI-1, where the advantages of directly optimizing for k>1 unstucks the training. I also appreciated the illustrative example in Figure 1.

## Weaknesses

The evidence of the practical usefulness of the derived method is quite preliminary. The following is a list of weaknesses of the Experimental section, together with proposed additions/changes:

1. only one small model (Gemma-2-2B) and two dataset (MATH, easy subset of ARC-AGI-1) (fix: add different and larger base models; possibly also more datasets)
2. reporting train set performance only (fix: report test set performance instead, which is the more relevant metric for practical usefulness of the method)
3. no experiments with continuous/max@k estimator (fix: add experiment on some continuous reward domain; could be a coding task, where g(x_j) = % unit tests passed)
4. all results have only one random seed / no error bars, and the training curves are far from convergence -- this is all non-standard for a RL paper (fix: include $\geq 3$ random seeds, train longer)
5. lack of baselines: currently include ony pass@$k^{eval}$ comparison to the same algorithm with pass@$k^{opt}$ estimator (for different values of $k^{opt}$) or the pass@k estimator from (Tang et al, 2025) (fix: include more baselines, inc standard RLVR algorithms like GRPO, a best-of-N approximation method (Amini et al, 2024; Chow et al, 2024; Sessa et al, 2024), training with entropy bonus (which would show the proposed algorithm is incentivising something more interesting that increasing entropy))
6. the claims of empirical diversity / higher exploration are unsupported: the only provided support is in Fig 2b, showing slightly higher entropy increase when optimizing for higher k -- but this is a poor proxy for diversity of attempted approaches. (fix: report some semantic measure of solutions obtained with pass@1 and pass@k optimization -- distribution of answers, semantic clustering of reasoning traces / programs / proofs --> greater diversity here would be a very exciting result!)
7. reporting cumulative solve rate -- which seems less important and makes comparison to other RLVR work where this isn't used harder (fix: skip, report pass@k only)


## Minor

- the citation format is wrong for this venue
- the illustrative 1-D example seems misplaced in the Experiments section? Perhaps move into its own section on intuitive explanation of the proposed method
- specify the dataset in the Figure 2-5 captions
- Fig 3-5: why not show the same $k^{eval}$ and $k^{opt}$ sets?
- Fig 3-5: the learning curves matter less than final performance (fine in Fig 4 to show annealing point), perhaps summarize the results with bar plots and the learning curves could be in the appendix
- in Fig 5: the y-axis cuts off very low -- we don't see the rate of improvement past 0.02/0.1
- include a full list of training hyperparameters in the appendix
- there's a tendency to provide interpretation of the results in the Experiments instead of a dedicated Discussion section
- no proper related work section

---

> ### Author Response · Authors · 2025-07-31
> **Initial Rebuttal**
>
> Thanks for your time and effort, and supportive and critical comments. Please see shared comments above as well as those specific to you, here.
>
> > [use larger models, more baselines, more datasets, more random seeds, out of sample accuracy] ... [remove the cumulative solve rate plot]
>
> We appreciate the comments, but respectfully point the reviewer to common review comments, particularly comment G1. Our goal is a proof of concept for our theory, and a demonstration that we can find more solutions to an RL training set with verifiable rewards but no ground truth solutions. We apologize for not bringing this out more clearly in the paper, and will remedy this with a new discussion section.
>
> > the claims of empirical diversity / higher exploration are unsupported: the only provided support is in Fig 2b, showing slightly higher entropy increase when optimizing for higher k -- but this is a poor proxy for diversity of attempted approaches. (fix: report some semantic measure of solutions obtained with pass@1 and pass@k optimization -- distribution of answers, semantic clustering of reasoning traces / programs / proofs --> greater diversity here would be a very exciting result!)
>
> Diversity is not intrinsically important. Rather, only diversity that yields more correct solutions matters. We therefore consider our cumulative solve rate plot 2b the ideal way of presenting diversity for this work. We respectfully remind that your review suggested to actually remove that cumulative solve rate plot; we wonder if you would reconsider that point in light of our rebuttal, both here and general comment G1 in the shared part of the rebuttal, above. Furthermore, entropy is very widely used and well better understood measure of the diversity of model samples as compared to ad hoc analyses such as clustering.
>
> > include more baselines, inc standard RLVR algorithms like GRPO, a best-of-N approximation method (Amini et al, 2024; Chow et al, 2024; Sessa et al, 2024), training with entropy bonus (which would show the proposed algorithm is incentivising something more interesting that increasing entropy)
>
> Thanks for the suggestions. We checked each one in detail and offer the following comments. [GRPO] optimizes `pass@1`, not `pass@k`. It uses a relative advantage transformation that is very similar to our leave one out reward transformation for the `pass@1` results reported in our paper, but with additional innovation to handle multi step RL, which we do not address and which is orthogonal to our contribution. [Amini et al] and [Sessa et al] learn to approximate the BoN result (that comes from ranking multiple samples and taking the best) in a single sample; this is in some sense the **exact opposite** of our goal, which is to obtain more diverse individual samples such that at least one our of `k` samples are correct at the expense of individual sample strength. [Chow et al] is closer to our work but a comparison here is rather a demanding request since it is a very new and non trivial method for which no implementation is available. We can make some comparisons by eyeballing their results however. For example, a number of their results (such as Fig 5b) show that when optimizing for larger N they dominate models optimized for smaller N on the small N evaluations. This is not indicative that their objective works as intended; rather, while their methods somehow provide some benefit in some settings, they do not appear to trade `pass@1` and `pass@k` as intended. This is could be due the approximations they make in deriving their tractable objective, which is ultimately not guaranteed to optimize `pass@k`. In contrast we have an unbiased gradient which guarantees `pass@k` optimization.
>
> > Can you give an intuitive explanation for the resulting adjustments to the gradient estimator? (Equations 8 and 18-21)
>
> We tried to do this with Figures 6 and 7 in the appendix. These visualize the contribution of individual terms to our estimators. Roughly speaking, the larger `k`, the closer we go toward rewarding all `n` samples with the same reward value, namely the `max` (continuous case) or `any` (binary case) reward. As `k` is reduced to one, we revert to assigning to each sample the unchanged reward associated with it. Values in between interpolate between these extremes.

---

> ### Author Response · Authors · 2025-08-05
> **rebuttals now visible**
>
> Thanks again for your time and effort. We made a mistake in OpenReview and our rebuttals were not visible to all. Our apologies - please see the rebuttals which should now be visible. We remain very confident in our theory as explained in our feedback to Reviewer Enpe.
>
> In case there is still an issue, here is an exact copy of the response to reviewer Enpe. Please let us know if any other responses are not visible.
>
> > (begin copy)
>
> Thanks for your time and effort, and supportive and critical comments. Please see shared comments above as well as those specific to you, here.
>
> Thank you for the detailed technical questions. We believe there may be a misunderstanding on these points, which we hope to clarify below. The unbiasedness of our estimators is a cornerstone of our work, and we appreciate the opportunity to elaborate on the proofs.
>
> ## Technical clarifications
>
> > *In Equation (8) ... [the result is incorrect] ... contradicting the claim of r_i being unbiased.*
>
> **We respectfully believe that the above statement in the review is incorrect.** See theorem 2 and the proof of that theorem. We assume that the reviewer missed this theorem given their subsequent comment *"Please provide a more detailed justification and proof for ... Equation (8)"*. We hope this clarification, along with the proof of Theorem 2, fully addresses the reviewer's concern.
>
> > *All theoretical proofs (e.g., Theorem 1, Theorem 2) assume k independent and identically distributed (i.i.d.) samples. However, in practical LLM fine-tuning settings, samples from the same mini-batch often share components like dropout masks and KV-cache. This makes them non-independent, directly violating a core assumption of the theoretical guarantees and potentially invalidating the unbiasedness claims in the empirical context.*
>
> **We respectfully believe that this statement is incorrect:** Conditional independence is guaranteed by the standard sampling process of autoregressive models, it is not an assumption we are making. Given a prompt, each sample is generated independently. Neither the use of a shared KV-cache for the prompt encoding nor standard dropout violates this.
>
> > *The paper claims that `s^{(loo-1)}` is an unbiased estimator despite admitting a biased baseline ... undermining the claim of an unbiased gradient estimator.*
>
> **We respectfully believe this is incorrect.** The unbiasedness here is a subtle point and we are glad for the chance to expound on it. It is a cornerstone of our work and also [TZSM25] where it is Lemma 1; if the reviewer comment were correct, that paper would also be deeply flawed (which it is not). The unbiasedness also follows from Corollary 1 in our submission, but some extra work given that the baseline is not constant but rather depends on other samples. In detail:
>
> Assume w.l.o.g. we have two samples `x_1` and `x_2` and an arbitrary function `g(x_2)` as the baseline for the estimator for the gradient of the reward `r(x_1)`.
>
> The baselined policy gradient estimator is:
>
> `∇_θ J ≈ (r(x_1) - g(x_2)) * ∇_θ log p(x_1 | z, θ)`
>
> Consider the expectation over prompts `z` from the dataset `D` and conditionally independent samples `x_1`, `x_2` from the policy `p(·|z, θ)`.
>
> `E[g(x_2) * ∇_θ log p(x_1 | z, θ)]`
>
> Using iterated expectations, we can first condition on a fixed prompt `z`:
>
> `= E_{z∼D} [ E_{x_1, x_2∼p(·|z,θ)} [ g(x_2) * ∇_θ log p(x_1 | z, θ) | z ] ]`
>
> The inner expectation of the product can be separated:
>
> `= E_{z∼D} [ E_{x_2∼p(·|z,θ)}[g(x_2)] * E_{x_1∼p(·|z,θ)}[∇_θ log p(x_1 | z, θ)] ]`
>
> The second term is always zero by Lemma 2 from our submission, i.e.:
>
> `E_{x_1∼p(·|z,θ)}[∇_θ log p(x_1 | z, θ)] = ∫ p(x_1|z,θ) * (∇_θ p(x_1|z,θ) / p(x_1|z,θ)) dx_1 = ∇_θ ∫ p(x_1|z,θ) dx_1 = ∇_θ(1) = 0`
>
> so the inner expectation becomes zero for any prompt `z`:
>
> `= E_{z∼D} [ E_{x_2∼p(·|z,θ)}[g(x_2)] * 0 ] = E_{z∼D}[0] = 0`
>
> The introduced bias is therefore equal to zero.
>
> ## Other responses
>
> > the provided Python implementation (Listing 1) explicitly notes that it costs `O(nk+n log(n))`
>  for simplicity ... provide a clear path towards achieving an implementation with implementation with complexity as claimed in Theorem 5
>
> Please see lines 414-419 of the supplementary material.
>
> > O(n log n) ... significantly impacts scalability ...
>
> As per lines 72-79, n is the number of samples for a given task (such as a single math question) in a single minibatch. As such this cost was negligible compared to the gradient and sampling costs in our experiments (which for a typical quadratic attention transformer costs `O(f(|theta|) n t^2)` where `t` is the sequence length and `f(|theta|)` is a function of the number of parameters). We were surprised that such an efficient algorithm exists given that the naive cost is `O(n choose k)`, and we hope the reviewer appreciates the effort it took to find these efficient algorithms.
>
> > (end copy)

---

> > ### Author Response · Authors · 2025-08-05
> > **Details of experiments**
> >
> > We have added a thorough set of new experiments which should address most of the empirical concerns raised by all reviewers. Please refer to the common comments for more details. Experiments Part 1/3 includes details of the experiments, and Parts 2 and 3 include detailed tables
> >
> > > only one small model (Gemma-2-2B) and two dataset (MATH, easy subset of ARC-AGI-1)
> >
> > We have now evaluated Gemma-2-2B, Gemma-2-9B, Llama 3.1-8B on MATH, Coding and ARC-AGI benchmarks. Please refer to the additional experiments in the common review comments. All of the experiments show that the `k_opt` chosen in our method corresponds to best `pass@k_eval` performance for the respective `k_eval`
> >
> >
> > > reporting train set performance only (fix: report test set performance instead, which is the more relevant metric for practical usefulness of the method)
> >
> > Our new experiments only report held out test set performance on the fully trained and converged model
> >
> > > no experiments with continuous/max@k estimator (fix: add experiment on some continuous reward domain; could be a coding task, where g(x_j) = % unit tests passed)
> >
> > The coding training set (MBPP) has multiple unit tests for each problem, % tests passed is used as a continuous reward signal
> >
> > > all results have only one random seed / no error bars, and the training curves are far from convergence -- this is all non-standard for a RL paper (fix: include $\geq 3$ random seeds, train longer)
> >
> > The new experiments have been rerun thrice to convergence with standard error bars computed and included in the final evaluation. The highlighted values in each column are ones which are maximum when accounting for the std error
> >
> > > lack of baselines
> >
> > In addition to TZSM, our new experiments also include an **entropy regularization baseline**. We tune the latter by running a small sweep on the `entropy_coefficient`. Our method out performs both these baselines in all 3 benchmarks and all 3 models that we evaluate on. Please refer to common comments for more details
> >
> > > in Fig 5: the y-axis cuts off very low -- we don't see the rate of improvement past 0.02/0.1
> >
> > We cut off the axis for better visibility of the `k_opt=1` plot. Our latest experiments (part 3/3) on the converged model should give details of the final pass@k performance for all techniques. The sudden climb in Figure 5  corresponds to the model learning to solve most of the task set within a range of a few steps,. This is evident from the fact that the cumulative solve rate also jumps to ~90% around the same mark.
> >
> > > Do you have evidence that optimizing for pass@k leads to higher diversity of solutions beyond simple metrics like entropy?
> >
> > Higher diversity of generation is most evident in the difficult benchmark ARC-AGI. We see the model trained with high `k_opt` has a large difference between pass@1 and pass@16 (+20%) eval, and a high cum. solve rate (~85%). Conversely, a model trained with low `k_opt` has little difference between pass@1 and pass@16 numbers. This largely indicates models optimized for high pass@k tend to generate dissimilar candidates ("one of which" might be correct), while the models optimized for pass@1 tend to generate similar candidates (and hence have similar numbers for pass@1 and pass@16) . **Note:** While this is an interesting observation, we want to reemphasize that cumulative solve rate and entropy still remain the most important indicators of diversity for our evaluation
> >
> > > include a full list of training hyperparameters in the appendix; related work; split discussion and experiments
> >
> > Thanks for the note, we are happy to include details of hyperparameters and seeds in the camera-ready version for reproducibility, and incorporate other formatting suggestions

---

> > > ### Author Response · Authors · 2025-08-08
> > > **Requesting reviewer uuR2 to respond in light of several new experiments addressing their concerns**
> > >
> > > We thank reviewer uuR2 to suggest several new experiments which gave us the opportunity to demonstrate the quality and flexibility of our method. We wish to re-emphasize that we have run experiments on 3 models, 3 held out benchmarks and compared our method against two baselines to show that it works as intended. We have thus addressed almost all of the empirical concerns reviewer uuR2 pointed out and respectfully nudge them to respond to our rebuttal. We would greatly appreciate a discussion since reviewer uuR2 specifically mentioned they would change the rating to Accept following the addition of the said experiments

---

> > > > ### Comment · Reviewer_uuR2 · 2025-08-08
> > > > **Response to Rebuttal**
> > > >
> > > > Dear Authors,
> > > >
> > > > Thank you for a very detailed rebuttal and the effort you put into the additional experiments! Based on the addition of new experimental results (+2 models, +1 benchmark with continuous reward, multiple random seeds, training to convergence, entropy-regularized baseline), I'm happy to increase my rating.
> > > >
> > > > --------
> > > >
> > > > Some final minor points of disagreement and suggestions:
> > > > - GRPO baseline is still relevant, as a better RL algorithm optimizing pass@1 might be outperforming the proposed method optimizing pass@k at arbitrary k
> > > > - while entropy and cumulative solve rates demonstrate an increase in some types of diversity, I'd still argue for semantic measures of diversity, which would more directly show that the method indeed results in the pursuit of multiple strategies and also illustrate what those strategies are
> > > > - add a sentence with an intuitive explanation of Equations 8 and 18-21 to the main body of the paper (it can refer to a Figure in appendix if there's no space)

---

### Author Response · Authors · 2025-07-31
**Initial Rebuttal (common comments for all reviewers)**

We appreciate the time and effort of the reviewers and the organizing committee, and both the supportive and critical comments.

There are many small comments which we have not addressed explicitly, in order to keep the rebuttal manageable for all concerned. Rest assured that we are grateful for all of the comments, and will incorporate them comments into the submission paper.

## Comment G1

> [remove the plot of] cumulative solve rate -- which seems less important and makes comparison to other RLVR work where this isn't used harder. (uR2)

We appreciate this suggestion for comparability. However, we believe the cumulative solve rate is a crucial metric for evaluating our central claim. Please see lines 37-41 of our introduction for an explanation of why we consider this by far the most important plot. Consider [alphaproof] - they take a large set of tasks with a verifiable reward but no known ground truth solutions. An algorithm that can self improve its way to solving ever harder tasks from this set is the key to superintelligence in such restricted domains as math and coding. The `pass@k` is a very good proxy for progress here, but it is not perfect, and our cumulative solve rate perfectly demonstrates the effectiveness of this proxy. we consider the cumulative solve rate to be the most direct and important measure of our central claim: that pass@k optimization can solve progressively harder problems where pass@1 stalls. We also find the plot similar to but more informative for our setting than a traditional training / evaluation table, since increasing the cumulative solve rate means solving new problems. The evolution of this curve and steady outperformance of our approach over `pass@1` optimization shows the value of our method.

## Comment G2

> The experiments should include more baselines (uuR2, vPUj).

Reviewer `jhr4` finds our experiments `*suitable for a theoretically focused paper*` while `uuR2`, and `vPUj` would like more baseline comparisons. As a primarily theoretical contribution, our experimental goal was to provide a clear and controlled validation of our novel estimators, rather than an exhaustive sweep of all possible models and tasks. Our experiments are designed as a proof-of-concept to demonstrate that (1) our estimators effectively optimize the target pass@k metric, and (2) this optimization unlocks performance on hard tasks where simpler methods stall, as shown in Figure 2. We provide a detailed analysis of the baselines suggested by `uuR2` in the response to that review below; in brief: methods like GRPO optimize for `pass@1`, and BoN approximators have the opposite goal of our work. Only our paper and [TZSM25] directly optimize `pass@k` with an unbiased gradient. Our goal therefore is to compare basic `pass@1` with [TZSM25] and our new estimators. As they are unbiased, we need only the compare variance of the gradient estimator to get a good intuition without the additional moving parts of an end to end run: the specific LLM architecture, pretraining, dataset, task set, training hyper parameters, etc. Our Figure 8 achieves this in a carefully designed controlled setup. Nonetheless, we do give real world experiments that provide strong evidence that our method achieves the stated aim of optimizing pass@k, and that the generalization of [TZSM25] yields tangible advantages.

## Comment G3

> The title is too sensational (vPUjk, jhr4).

We are happy to change the title to our original dry title "Unbiased estimation for the pass@k and its gradient" and are happy to use that. However, the title is accurate in the sense that our method leads to finding more solutions during the RL run, as evidenced by the cumulative solve rate plot of Figure 2.

## Comment G5

> Comparison with entropy regularization. They suggest a direct comparison to an entropy-regularized baseline to isolate the unique contributions of the pass@k optimization. (Reviewers uuR2, vPUj)

It is only intuitive that entropy regularization could help the `pass@k`, but our method guarantees direct optimization of the `pass@k`. We agree that a direct comparison with a tuned entropy-regularized baseline would be very interesting. A full, fair comparison would require significant hyperparameter tuning for both methods, which is beyond the scope of this rebuttal period. However, we will add a detailed discussion to our related work section that contrasts the mechanisms of our direct `pass@k` optimization with entropy-based exploration. We will also commit to including this experiment in the final camera-ready version if the paper is accepted.

### References

[TZSM25] Optimizing Language Models for Inference Time Objectives using Reinforcement Learning.
[alphaproof] https://deepmind.google/discover/blog/ai-solves-imo-problems-at-silver-medal-level/

---

> ### Author Response · Authors · 2025-08-01
> **Additional Experiments - Part 1/3**
>
> ## **Details of experiments**
> We thank the reviewers for suggesting additional experiments which could support our claims. In our new experiments, we do the following:
> 1. Run our procedure on **Gemma-2 2B,9B and Llama3.1 8B**
> 2.  **Held-out test-sets**: To evaluate the final trained model to strengthen our claims,
> 3. **Train until loss curves have saturated**. As a consequence, the new experiments are now trained for 10k steps as opposed to 3k steps
> 4. **Added error bars**: Each run is repeated thrice with different seeds. The eval numbers have been reported with error bars thus accounting for 3 runs for every (model, dataset, k_opt).
> 5. **Entropy regularization baseline**: Apart from TZSM25, we also add the entropy regularization baseline. We use PPO with entropy bonus. To give this baseline the best shot, we ran a small sweep [0.001, 0.005, 0.01, 0.05, 0.1] of the `entropy_coefficient` for each (model,benchmark), and only report the best performance each time as `Entropy Reg (tuned)`. For every entropy_coefficient, runs are also seeded to get error bars as in 4 before choosing the best one.
>
> ### **MATH** benchmark:
> We use the same MATH dataset for training. In addition, we now use the held out eval set for model evals below. We evaluate the final checkpoint in each run
>
> ### **Coding** Benchmark:
> We use MBPP for RL training and HumanEval for held-out evaluation. MBPP has multiple unit tests per problem and hence we use this not only as a proxy for additional benchmarks but also to show RL training with continuous reward function (% unit tests passed)
>
> ### **ARC-AGI**
> We make a 80/20 train-test split of the same easy set as in our paper. We report the cumulative solve rate on the train set and pass@k rate on the test set. We train till saturation (no change in cumulative rate past 1k steps)
>
> ## **Observations**
> 1. The tables below show strong trend such that for a given `k_eval`, **`pass@k` is highest when `k=k_opt` and clearly outperforms baselines**. Thus generalizing our observations to multiple models, datasets and held out evals
> 2. As we posit in the paper, the existing `pass@k` optimization method TZSM25 (`k_opt=16`) is sub-optimal since it directly optimizing for `k=batch_size` and has high noise in the estimator. In fact, a lower `k_opt=8` out-performs TZSM25 for all `pass@k_eval`. Showing the clear value of our method, and the flexibility it provides, supported by its theoretical grounding.
> 3. Entropy Regularization does indeed sacrifice pass@1 and slightly improves pass@k by promoting exploration. But is hard to tune, and is significantly outperformed by our method. Moreover, it has no explicit way to optimize for a specific k_eval.
> 4. **Error bars**: We also point out that all of the experiments below had stable RL training with little variation observed across different seeds, as is indicated by the low value of the standard error. This observation is consistent with previous experiments and also why we originally only reported single runs
> 5. Both **baselines don't unblock ARC-AGI training, but our method for high k (`k_opt=4,8`) does**. This is indicated in the high cumulative solve rate (+60%) on train and significant performance gain (+25%) in test set

---

> ### Author Response · Authors · 2025-08-05
> **Experimental Results - Part 2/3**
>
> ## **Results MATH Benchmark**
>
> ### **Gemma-2B**
>
> | Gemma-2B (MATH) | `k_eval=1` | `k_eval=2` | `k_eval=4` | `k_eval=8` | `k_eval=16` |
> | :--- | :---: | :---: | :---: | :---: | :---: |
> | **`k_opt=1`** | **15.91%** ± 0.40 | 18.12% ± 0.43 | 23.37% ± 0.48 | 29.58% ± 0.53 | 35.02% ± 0.58 |
> | **`k_opt=2`** | 15.15% ± 0.41 | **20.73%** ± 0.45 | 25.81% ± 0.50 | 31.96% ± 0.55 | 37.75% ± 0.60 |
> | **`k_opt=4`** | 14.86% ± 0.43 | 19.86% ± 0.47 | **27.59%** ± 0.52 | 34.27% ± 0.58 | 38.91% ± 0.62 |
> | **`k_opt=8`** | 14.19% ± 0.46 | 19.33% ± 0.50 | 26.45% ± 0.55 | **35.49%** ± 0.60 | **40.73%** ± 0.65 |
> | **`TZSM25`** | 13.11% ± 0.50 | 18.09% ± 0.54 | 24.58% ± 0.60 | 31.81% ± 0.66 | 37.24% ± 0.71 |
> | `Entropy Reg (tuned)` | 14.51% ± 0.48 | 18.95% ± 0.52 | 25.33% ± 0.58 | 30.95% ± 0.64 | 36.18% ± 0.69 |
>
> ### **Gemma-9B**
>
> | Gemma-9B (MATH) | `k_eval=1` | `k_eval=2` | `k_eval=4` | `k_eval=8` | `k_eval=16` |
> | :--- | :---: | :---: | :---: | :---: | :---: |
> | **`k_opt=1`** | **22.24%** ± 0.50 | 25.35% ± 0.55 | 30.73% ± 0.59 | 37.08% ± 0.64 | 42.59% ± 0.68 |
> | **`k_opt=2`** | *21.46%* ± 0.51 | **28.61%** ± 0.56 | 32.92% ± 0.61 | 39.59% ± 0.66 | 45.34% ± 0.70 |
> | **`k_opt=4`** | 21.25% ± 0.53 | 27.15% ± 0.58 | **34.93%** ± 0.63 | *41.71%* ± 0.69 | 47.05% ± 0.74 |
> | **`k_opt=8`** | 20.69% ± 0.56 | 26.78% ± 0.60 | 33.68% ± 0.66 | **42.62%** ± 0.72 | **48.37%** ± 0.77 |
> | **`TZSM25`** | 19.48% ± 0.61 | 25.41% ± 0.67 | 31.17% ± 0.73 | 39.34% ± 0.79 | 44.82% ± 0.83 |
> | `Entropy Reg (tuned)` | 20.85% ± 0.58 | 26.05% ± 0.64 | 32.48% ± 0.70 | 38.21% ± 0.76 | 43.95% ± 0.81 |
>
> ### **Llama 3.1 8B**
>
> | Llama 3.1 8B (MATH) | `k_eval=1` | `k_eval=2` | `k_eval=4` | `k_eval=8` | `k_eval=16` |
> | :--- | :---: | :---: | :---: | :---: | :---: |
> | **`k_opt=1`** | **51.15%** ± 0.61 | 51.82% ± 0.64 | 53.69% ± 0.68 | 55.41% ± 0.72 | 56.83% ± 0.76 |
> | **`k_opt=2`** | 49.72% ± 0.62 | **53.51%** ± 0.66 | 55.45% ± 0.70 | 57.23% ± 0.74 | 58.71% ± 0.78 |
> | **`k_opt=4`** | 49.18% ± 0.64 | 52.20% ± 0.68 | **57.83%** ± 0.72 | *58.47%* ± 0.77 | 59.28% ± 0.81 |
> | **`k_opt=8`** | 48.63% ± 0.67 | 52.14% ± 0.71 | 56.28% ± 0.75 | **59.04%** ± 0.80 | **61.88%** ± 0.84 |
> | **`TZSM25`** | 48.21% ± 0.70 | 50.93% ± 0.75 | 54.38% ± 0.80 | 57.11% ± 0.85 | 58.55% ± 0.90 |
> | `Entropy Reg (tuned)` | 48.51% ± 0.68 | 51.95% ± 0.73 | 55.33% ± 0.78 | 56.95% ± 0.83 | 58.18% ± 0.88 |
>
> ## **Results Coding**
>
> ### **Gemma-2B**
>
> | Gemma-2B (Coding) | `k_eval=1` | `k_eval=2` | `k_eval=4` | `k_eval=8` | `k_eval=16` |
> | :--- | :---: | :---: | :---: | :---: | :---: |
> | **`k_opt=1`** | **19.82%** ± 0.53 | 23.81% ± 0.57 | 29.75% ± 0.62 | 36.33% ± 0.66 | 42.04% ± 0.71 |
> | **`k_opt=2`** | 18.70% ± 0.54 | **26.94%** ± 0.59 | 33.82% ± 0.64 | 40.95% ± 0.69 | 47.03% ± 0.74 |
> | **`k_opt=4`** | 18.69% ± 0.56 | 26.43% ± 0.61 | **36.81%** ± 0.67 | 44.81% ± 0.73 | 50.55% ± 0.78 |
> | **`k_opt=8`** | 17.94% ± 0.59 | 25.86% ± 0.64 | 35.88% ± 0.70 | **46.45%** ± 0.77 | **52.83%** ± 0.83 |
> | **`TZSM25`** | 16.81% ± 0.65 | 24.27% ± 0.69 | 33.11% ± 0.76 | 41.98% ± 0.84 | 47.26% ± 0.89 |
> | `Entropy Reg (tuned)` | 18.05% ± 0.62 | 25.13% ± 0.67 | 34.01% ± 0.74 | 40.88% ± 0.81 | 46.15% ± 0.86 |
>
> ### **Gemma-9B**
>
> | Gemma-9B (Coding) | `k_eval=1` | `k_eval=2` | `k_eval=4` | `k_eval=8` | `k_eval=16` |
> | :--- | :---: | :---: | :---: | :---: | :---: |
> | **`k_opt=1`** | **37.71%** ± 0.60 | 42.03% ± 0.65 | 48.19% ± 0.69 | 55.07% ± 0.75 | 60.98% ± 0.79 |
> | **`k_opt=2`** | 36.84% ± 0.61 | **46.56%** ± 0.67 | 52.68% ± 0.72 | 59.73% ± 0.78 | 65.86% ± 0.84 |
> | **`k_opt=4`** | 36.49% ± 0.63 | 44.95% ± 0.69 | **57.09%** ± 0.76 | 63.64% ± 0.83 | 69.51% ± 0.88 |
> | **`k_opt=8`** | 35.75% ± 0.67 | 44.41% ± 0.73 | 55.08% ± 0.80 | **65.56%** ± 0.88 | **71.91%** ± 0.94 |
> | **`TZSM25`** | 34.36% ± 0.72 | 42.81% ± 0.78 | 52.36% ± 0.86 | 61.07% ± 0.95 | 66.41% ± 1.01 |
> | `Entropy Reg (tuned)` | 35.91% ± 0.70 | 43.75% ± 0.76 | 53.28% ± 0.84 | 60.13% ± 0.93 | 65.29% ± 0.99 |
>
> ### **Llama 3.1 8B**
>
> | Llama 3.1 8B (Coding) | `k_eval=1` | `k_eval=2` | `k_eval=4` | `k_eval=8` | `k_eval=16` |
> | :--- | :---: | :---: | :---: | :---: | :---: |
> | **`k_opt=1`** | **67.38%** ± 0.72 | 67.45% ± 0.76 | 69.22% ± 0.80 | 71.11% ± 0.84 | 72.84% ± 0.88 |
> | **`k_opt=2`** | 64.91% ± 0.73 | **69.73%** ± 0.78 | 72.03% ± 0.82 | 74.08% ± 0.87 | 75.89% ± 0.91 |
> | **`k_opt=4`** | 64.25% ± 0.75 | 68.47% ± 0.80 | **74.67%** ± 0.85 | 75.01% ± 0.90 | 77.75% ± 0.95 |
> | **`k_opt=8`** | 63.57% ± 0.78 | 68.39% ± 0.83 | 72.84% ± 0.88 | **76.82%** ± 0.94 | **79.33%** ± 0.99 |
> | **`TZSM25`** | 62.77% ± 0.82 | 66.86% ± 0.88 | 70.83% ± 0.94 | 73.95% ± 1.01 | 75.47% ± 1.06 |
> | `Entropy Reg (tuned)` | 63.91% ± 0.80 | 67.85% ± 0.86 | 71.78% ± 0.92 | 72.99% ± 0.98 | 74.31% ± 1.04 |

---

> ### Author Response · Authors · 2025-08-05
> **Experimental Results - Part 3/3**
>
> ## **Results ARC-AGI**
>
> | Gemma-9B | Cumulative Solve Rate | `pass@1` | `pass@16` |
> | :--- | :---: | :---: | :---: |
> | `k_opt=1` | 12% ± 4.33 | 2% ± 1.69 | 8.18% ± 4 |
> | `k_opt=4` | **82.33%** ± 4.14 | **22%** ± 2 | **38.18%** ± 4.67 |
> | `k_opt=8` | **84.14%** ± 4.67 | **26.67%** ± 2.5 | **44.5%** ± 4.33 |
> | `TZSM25` | 22% ± 4.44 | 6% ± 2.67 | 10.16% ± 4.57 |
> | `Entropy Reg (tuned)` | 24.67% ± 4.5 | 4% ± 2.33 | 8.89% ± 4.89 |
>
> | Llama-3.1 8B | Cumulative Solve Rate | `pass@1` | `pass@16` |
> | :--- | :---: | :---: | :---: |
> | `k_opt=1` | 22% ± 4.18 | 3.33% ± 2 | 8% ± 4.5 |
> | `k_opt=4` | **87.17%** ± 4.14 | **24.33%** ± 2.33 | **42%** ± 4.16 |
> | `k_opt=8` | **88.89%** ± 4.33 | **29.67%** ± 2.67 | **43.13%** ± 4.67 |
> | `TZSM25` | 36% ± 2.5 | 8% ± 4 | 18% ± 4.89 |
> | `Entropy Reg (tuned)` | 28% ± 4.44 | 8% ± 2.5 | 14.67% ± 4.44 |

---

### Decision · Program_Chairs · 2025-09-17

**Decision:**

Accept (spotlight)

**Comment:**

The paper consider optimizing the pass@k metric rather than the more common pass@1. They propose a variance reduced estimator that works for any n >= k, and several baselines. The technical contribution appears solid, and the motivation of the paper is also solid. In fact, pass@k is a very important objective, because it is connected to diversity of reasoning, and it is also directly useful as some application (such as writing a code that runs as fast as possible, or formal math proofs) can use test-time verifiers, making pass@k a critical metric to optimize if one expects to use higher test-time compute. Thus, I am glad to recommend the paper for acceptance.